# Modular synthesis of clickable peptides via late-stage maleimidation on C(7)-H tryptophan

Peng Wang [1,4], Jiang Liu [1,4], Xiaomei Zhu [1,4], Kenry [2,3], Zhengqing Yan [1], Jiahui Yan [1], Jitong Jiang [1], Manlin Fu [1], Jingyan Ge [1], Qing Zhu [1] ✉ & Yuguo Zheng[1]

Cyclic peptides have attracted tremendous attention in the pharmaceutical industry owing to their excellent cell penetrability, stability, thermostability, and drug-like properties. However, the currently available facile methodologies for creating such peptides are rather limited. Herein, we report an efficient and direct peptide cyclization via rhodium(III)-catalyzed C(7)-H maleimidation. Notably, this catalytical system has excellent regioselectivity and high tolerance of functional groups which enable late-stage cyclization of peptides. This architecture of cyclic peptides exhibits higher bioactivity than its parent linear peptides. Moreover, the Trp-substituted maleimide displays excellent reactivity toward Michael addition, indicating its potential as a click functional group for applications in chemical biology and medicinal chemistry. As a proof of principle, RGD-GFLG-DOX, which is a peptide-drug-conjugate, is constructed and it displays a strong binding affinity and high antiproliferative activity toward integrin-$\alpha v\beta_3$ overexpressed cancer cell lines. The proposed strategy for rapid preparation of stapled peptides would be a robust tool for creating peptide-drug conjugates.

Peptides have attracted increasing interests as an important class of bioactive molecules in proteomics research, pharmaceutical chemistry and biotechnology[1,2]. Particularly, non-natural and cyclic peptides derived from natural peptides often feature enhanced biological and pharmacokinetic properties than their matrix because of the higher stability against proteolysis, more stable peptide conformation, and longer effective time of drugs[3,4]. Moreover, cyclic peptides display excellent cell permeability and stronger binding affinity to protein surfaces that are involved in protein–protein interactions by lowering the energy barrier for adapting to the membrane environment and binding to transport proteins[5,6]. To access these cyclic peptides, canonical diversification strategies mainly proceed with amidation and esterification reactions at reactive side chains of amino acid residues,

such as lysine, tyrosine, aspartic acid and serine[7–9]. However, these methods cannot be widely used due to the excessive reliance on reactive functional groups of amino acid side chains, which may have important biological activities. Thus, highly efficient and site-specific functionalization of widespread C-H bond in peptides still remains a challenge[10].

Transition metal-catalyzed late-stage functionalization of complex molecules has a good site-specificity, high tolerance of functional groups, and therefore, providing a straightforward method for an efficient C-H functionalization/macrocyclization of peptides. This strategy provides distinct peptide scaffolds that are not easily achievable with traditional methods, which have been explored by Lavilla[11], Albericio[12], Shi[13,14], Ackermann[15–17], Wang[18–20], Yu[21], Chen[22,23]

[1]Key Laboratory of Bioorganic Synthesis of Zhejiang Province, College of Biotechnology and Bioengineering, Zhejiang University of Technology, Hangzhou 310014, China. [2]Harvard John A. Paulson School of Engineering and Applied Sciences, Harvard University, Allston, MA 02134, USA. [3]Department of Imaging, Dana-Farber Cancer Institute and Harvard Medical School, Boston, MA 02215, USA. [4]These authors contributed equally: Peng Wang, Jiang Liu, Xiaomei Zhu. ✉e-mail: zhuq@zjut.edu.cn

and others[24–27]. In this regard, the amino acid tryptophan (Trp) is an ideal modification residue due to its relatively low abundance in peptide/protein sequences and unique impact on biological events, which include a series of important natural products containing C(7)-substituted tryptophan (Fig. 1a and Supplementary Figure 1)[28–31]. Up to now, several modifications on Trp, including alkynylation[32], fluoroalkylation[33,34], arylation[35] and macrocyclization[36–40], have been successfully developed. For example, Liu et al.[40] successfully introduced maleimide into tryptophan and tryptophan-containing peptides at C(2) position by utilizing *N*-pyridine as directing group, and the

directing group and peptide protection groups of the functionalized peptides could be successfully removed (Supplementary Figure 2). However, these reactions are limited to the chemically activated C(2) position (Fig. 1b). Recently, Wang[41] presented the example of Pd-catalyzed C-H olefination of peptide macrocyclization at C(4) position of Trp promoted by an N-terminal trifluoromethanesulfonamide (Tf) group (Fig. 1c).

On the other hand, the Ackermann group developed the tryptophan C(7)-H amidation catalyzed by rhodium, which was facilitated by the directing group, *N*-pyridyl group[42]. Furthermore, Wang and Shi[43]

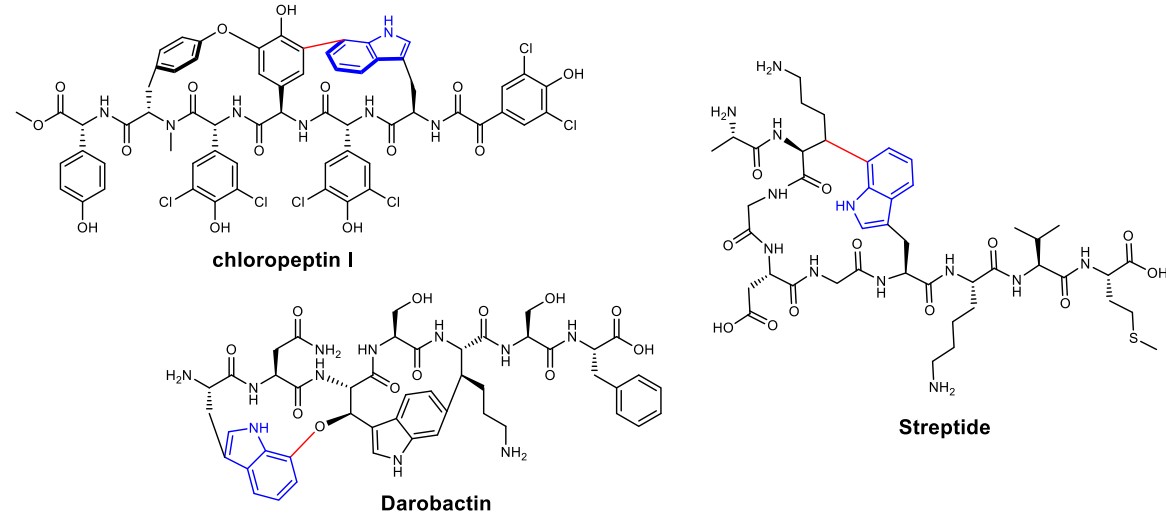

**a** Bioactive C7-cyclization Tryptophan-peptide Natural Products

**chloropeptin I**

**Darobactin**

**Streptide**

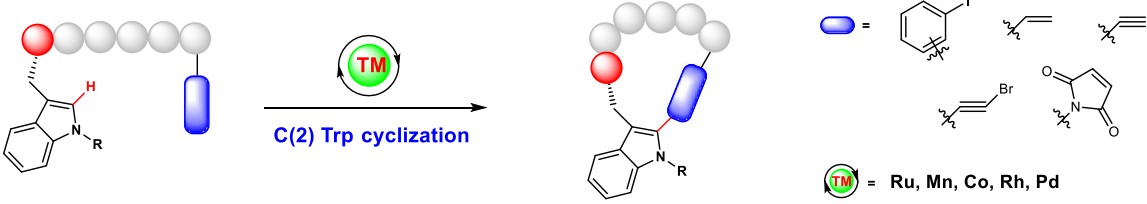

**b** Current major strategies for peptide macrocyclization on Trp:

C(2) Trp cyclization

TM = Ru, Mn, Co, Rh, Pd

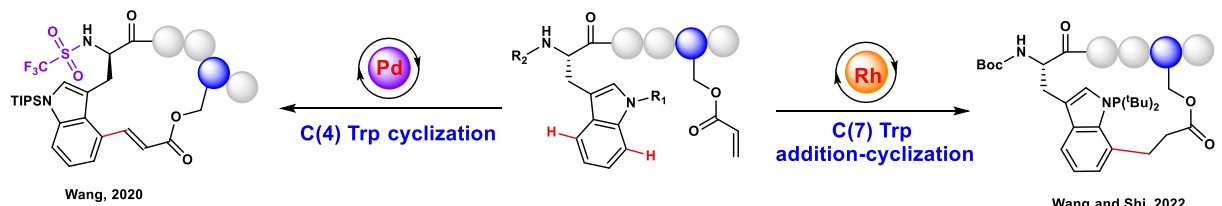

**c** Peptide macrocyclization of Trp residues at the C(4) and C(7) positions:

Wang, 2020

C(4) Trp cyclization

C(7) Trp addition-cyclization

Wang and Shi, 2022

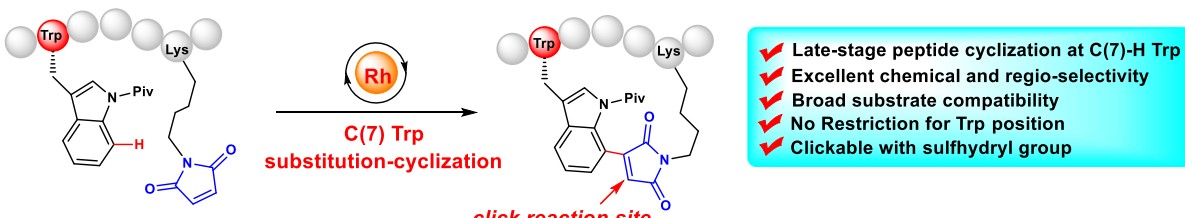

**d** This work:

C(7) Trp substitution-cyclization

*click reaction site*

✔ Late-stage peptide cyclization at C(7)-H Trp
✔ Excellent chemical and regio-selectivity
✔ Broad substrate compatibility
✔ No Restriction for Trp position
✔ Clickable with sulfhydryl group

**Fig. 1 | Direct C-H activation constructed stapled peptides in Trp residues. a** Bioactive C7-cyclization tryptophan-peptide natural products. Chemical tool box for peptide macrocyclization on Trp residue. Current strategies for peptide C-H macrocyclization on C(2)-H Trp (**b**), C(4)-H Trp and C(7)-H Trp (**c**); **d** Late-stage peptide substitution-cyclization via maleimidation on C(7)-H Trp in this work.

reported the late-stage peptide ligation and macrocyclization with olefins by Rh-catalyzed C(7)-H alkylation of Trp with a pre-installed *N*-P$^t$Bu$_2$ directing group, and found that the addition-cyclized products were strongly cytotoxic to cancer cells (Fig. 1c).

Maleimide and its derivatives play an important role in pharmaceutical chemistry and chemical biology[44,45]. They are not only important building blocks of drugs[46–48], but also, they can serve as a biocompatible and clickable handle for conjugating bioactive molecules via Michael addition[49,50], providing a powerful tool for the construction of antibody-drug conjugates (ADCs), including anticancer drug rentuximab vedotin and trastuzumab emtansine approved by FDA[51]. On the other hand, the pivaloyl group is an efficient directing group that has been used for C-H activation at unfavorable positions[52–54].

In this work, inspired by these findings, we are motivated to evaluate whether maleimide can be utilized to generate practical architectures of nonnatural/cyclic peptides via late-stage C–H functionalization on C(7) position of Trp facilitated by *N*-pivaloyl group (Fig. 1d). We anticipate that maleimide might contribute to the biological functions of this class of cyclic peptides.

## Results

### Optimization of reaction conditions

Rhodium(III)-catalyzed C(sp2)-H activation has been demonstrated as an efficient strategy for late-stage functionalization of complex compounds[55]. As such, we hypothesized that, by tuning metal catalysts, additives, and oxidant, and using *N*-pivaloyl as a directing group, the unique reactivity of C(7)-H Trp for C-H activation and its tolerance for functional groups could be increased to facilitate late-stage peptide functionalization.

To test our hypothesis, we explored the conditions for the rhodium(III)-catalyzed C(7)-H maleimidation reaction. Initially, Boc-Trp(Piv)-OMe (**1a**) and maleimide (**2a**) were used as the model substrates, which were catalyzed by [RhCp*Cl$_2$]$_2$ in the presence of Ag$_2$CO$_3$ and AgSbF$_6$ in THF, to offer **3a** and its oxidative product **3aa** in 41 and 13% yield, respectively, while product was obtained in the absence of AgSbF$_6$ (Table 1, **entrys 1** and **2**). This result inspired us to screen more silver additives for improving reaction yield and selectivity. As shown in Table 1, AgNTf$_2$ emerged as the best additive, and the yield was improved to 48% (**entry 4**). Among the solvents examined, DCM offered **3a** in 55% (**entries 5–7**). Further screening showed that transition-metal catalysts also had a significant influence on both the reaction selectivity and conversion rate (Table 1, **entries 8–10**). [IrCp*Cl$_2$]$_2$, [Ru(p-cymene)Cl$_2$]$_2$ or Pd(OAc)$_2$ remarkably lowered conversion to less than 15%. The effect of oxidants on the reaction was also explored (**entries 11–13**). Interestingly, when Ag$_2$O was used as an oxidant, **3a** was obtained in a yield as high as 83% and more importantly, no **3aa** was observed, suggesting the excellent selectivity. Increasing or decreasing the reaction temperature led to a decrease in the yield of **3a** (Table 1, **entrys 14** and **15**). Furthermore, no racemized products **3a** were found while using maleimide-decorated Boc-DL-Trp(Piv)-OMe as the internal standard (Supplementary Figure 6). Therefore, the optimum reaction condition was Boc-Trp(Piv)-OMe (**1a**) and maleimide (**2a**) in the presence of [RhCp*Cl$_2$]$_2$ catalyst, AgNTf$_2$ as additive and Ag$_2$O as oxidant in 1.5 mL dichloromethane.

### The maleimidylation of Trps and peptides

After establishing the optimal conditions, we next investigated the substrate scope and functional group tolerance (Fig. 2). First, the scope of protecting groups on Trp was examined. As shown in Fig. 2, traditional protecting groups, including *N*-Cbz, *N*-Ac, *N*-Boc, benzyl esters and ethyl esters, were compatible with the catalytic system (**3a–3f**, 76-83%). Then, we investigated the substrate scope of maleimides. Both alkyl and aryl substituents on the *N*-atom of maleimides were tolerated under the standard condition, and the

corresponding maleimidated Boc-Trp(Piv)-OMe was obtained in good to excellent yields (**3g–3n**, 55–80%). Notably, this optimized catalytic system successfully decorated maleimide to the C(7) site of tryptophan-derived drug melatonin (**3o**, 73%).

Next, the late-stage C(7)-H maleimidation of dipeptides was investigated (Fig. 3). First, dipeptides could be regioselectively maleimidated via C(7)-H activation in good yields, ranging from 62–82% (**5a-c**). Moreover, a dipeptide containing β-amino acid Boc-Trp(Piv)-β-Ala-OMe was efficiently transformed into the corresponding product **5d** in 79% yield. Under standard condition, more dipeptide types whose Trp was located at C-terminal (**4e–4h**) were directly decorated with maleimide in good yields (65–80%), suggesting the minimal effect of the position of Trp in peptides on the C-H activation reaction. Additionally, we explored the rhodium-catalyzed C(7)-H maleimidation of various unprotected dipeptides. Interestingly, the reaction system was compatible with several unprotected amino acid residues, such as aspartic acid (**4i**), asparagine (**4j$_1$**), and glutamine (**4j$_2$**), offering the corresponding products in moderate to good yields (**5i–5j**, 38%–71%). Nevertheless, it should be noted that the ε-amino group of lysine (**4k**) reacted with maleimide to give product **5k′** via the Michael addition reaction. The peptide **4l** containing cysteine failed to yield the desired product most probably due to the presence of the easily oxidized sulfhydryl group.

Encouraged by the successful maleimidation of dipeptides, we next explored the late-stage modification of Trp-containing complex peptides (Fig. 3). A wide range of tripeptides provided desired products with moderate to good yields (40%–75%) regardless of the location of Trp, whether it was located at the N-terminal (**5o** and **5p**), in the middle (**5 m** and **5n**), or at the C-terminal (**5q–5s**) of peptide chains. Furthermore, tetrapeptides Boc-Trp(Piv)-Asp(OtBu)-Gly-Val-OMe (**4t**) and Boc-Phe-Val-Gly-Trp(Piv)-OMe (**4 u**) offered the corresponding products **5t** (49%) and **5 u** (47%). More complex pentapeptide Ac-Phe-Lys(Boc)-Ile-Gly-Trp(Piv)-OMe (**4 v**) and Boc-Val-Ala-Leu-Gly-Trp(Piv)-OMe (**4w**) also adapted well to the catalytic system and the desired products **5 v** and **5w** could be obtained in 42% and 50%, respectively.

The regioselective ligation of complex substrates reflects the robustness of C-H activation manifold. *N*-amino acid/dipeptide-substituted maleimides were designed as substrates for C(7)-Trp ligation. As shown in Fig. 4, single amino acid and peptides were conjugated effectively to produce backbone of peptides **7a–7f** in 44%–73% yields. Notably, the complex peptides **7b** and **7 f** containing Ser and Thr were obtained in moderate yields. Moreover, under mild and non-racemic standard condition, amino acid **1a** and bioactive molecules (including natural product, sugar and fluorophores) containing maleimide were effectively hybridized to give **7g–7k** with yields ranging from 58% to 75%. The peptide **4j$_2$**, which contains an unprotected glutamine residue, can undergo maleimidation to generate glycopeptide **7k** in 65% yield. All of these results demostrated that this method could serve as a versatile tool for regioselective C(7)-H functionalizations of Trp-containing peptides.

### Cyclization of Trp-containing peptides

Motivated by the highly efficient maleimidation on C(7) of Trp, the possibility of cyclization by intramolecular Trp C(7)-H activation was then explored. Under the standard condition, the diluted **8a** (12 mM) offered the desired 22-membered macrocycle successfully, whose cyclic structure of **9a** was confirmed by X-ray analysis (Fig. 5a, Supplementary Table 1). To exploit the full potential of C(7) cyclization, we investigated the cyclization of peptides where Trp was placed at different positions. As shown in Fig. 5a, tripeptides and pentapeptides (**8b–8e**) with Trp at C-terminus were cyclized, providing 23-membered and 26-membered cyclic peptides **9b–9e** in moderate yields (15%–43%). Meanwhile, pentapeptide substrate (**8 f**) containing Trp at N-terminus produced 26-membered cyclic peptide **9 f** in 34% yield. Furthermore, hexapeptide (**8 g**), whose Trp was embedded in the

**Table 1 | Optimization of reaction conditions.[a]**

| entry | catalyst | additive | oxidant | solvent | 3a/3aa yield(%)[b] |
|---|---|---|---|---|---|
| 1 | [RhCp*Cl₂]₂ | | Ag₂CO₃ | THF | None/None |
| 2 | [RhCp*Cl₂]₂ | AgSbF₆ | Ag₂CO₃ | THF | 41/13 |
| 3 | [RhCp*Cl₂]₂ | AgBF₄ | Ag₂CO₃ | THF | 15/10 |
| 4 | [RhCp*Cl₂]₂ | AgNTf₂ | Ag₂CO₃ | THF | 48/15 |
| 5 | [RhCp*Cl₂]₂ | AgNTf₂ | Ag₂CO₃ | DCM | 55/20 |
| 6 | [RhCp*Cl₂]₂ | AgNTf₂ | Ag₂CO₃ | DMF | None/5 |
| 7 | [RhCp*Cl₂]₂ | AgNTf₂ | Ag₂CO₃ | DCE | 40/14 |
| 8 | [IrCp*Cl₂]₂ | AgNTf₂ | Ag₂CO₃ | DCM | 15/Trace |
| 9 | [Ru(p-cymene)Cl₂]₂ | AgNTf₂ | Ag₂CO₃ | DCM | Trace/Trace |
| 10 | Pd(OAc)₂ | AgNTf₂ | Ag₂CO₃ | DCM | None/None |
| 11 | [RhCp*Cl₂]₂ | AgNTf₂ | AgOAc | DCM | 27/12 |
| 12 | [RhCp*Cl₂]₂ | AgNTf₂ | Ag₂O | DCM | 83/Trace |
| 13 | [RhCp*Cl₂]₂ | AgNTf₂ | Cu(OAc)₂ | DCM | Trace/6 |
| 14[c] | [RhCp*Cl₂]₂ | AgNTf₂ | Ag₂O | DCM | 53/Trace |
| 15[d] | [RhCp*Cl₂]₂ | AgNTf₂ | Ag₂O | DCM | 78/Trace |

[a]Reaction conditions: 1a (0.2 mmol), 2a (0.6 mmol), catalyst (5 mol%), additive (20 mol%), oxidant (1.5 eq), solvent (1.5 mL), 80 °C, 6 h.
[b]Isolated yields.
[c]Temperature increased to 100 °C.
[d]Temperature decreased to 60 °C.

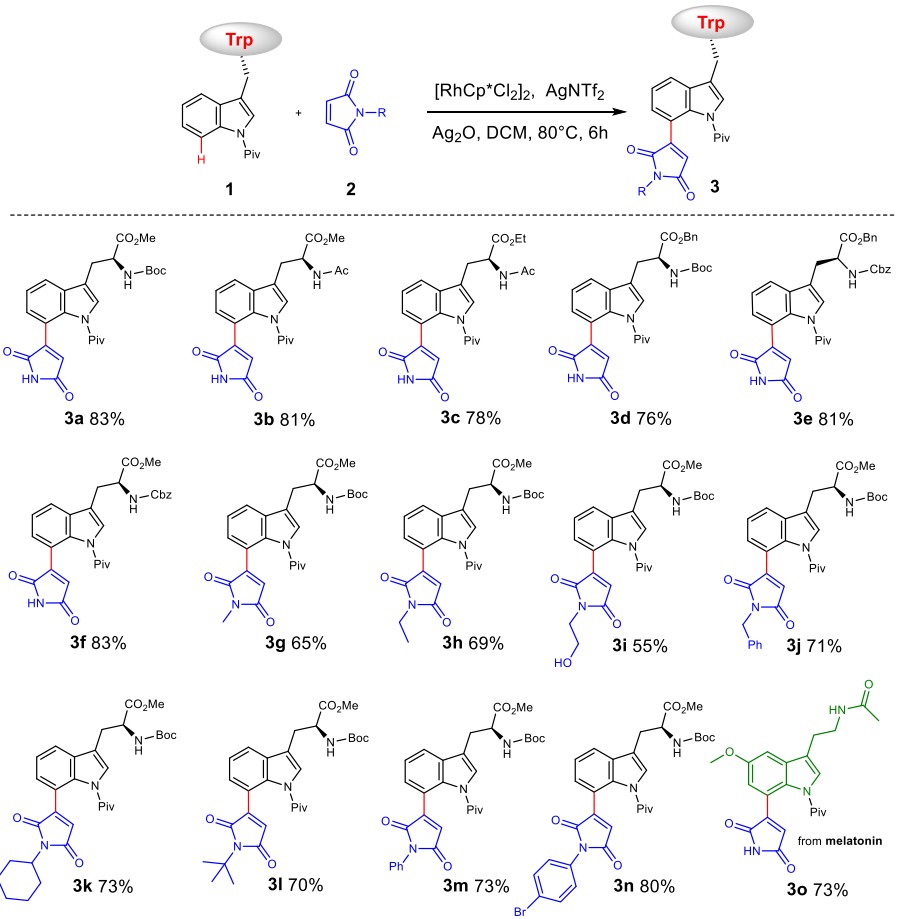

**Fig. 2 | Substrate scope for maleimidation of Trp or Trp-derivatives and maleimide derivatives.** Reaction conditions: **1** (0.20 mmol), **2** (0.60 mmol), [RhCp*Cl₂]₂ (5 mol %), AgNTf₂ (20 mol%), Ag₂O (1.5 eq), DCM (1.5 mL), 80 °C, 6 h.

middle of peptide, produced 26-membered stapled peptide **9 g** in 32% yield. In addition, 28-membered and 29-membered cyclic peptides (**9h**–**9j**) were obtained via the cyclization of peptide substrates **8h**–**8j** in moderate yields (32%–38%). A 29-membered cyclic arginine-glycine-aspartic acid (RGD) peptide (**9j**) was also successfuly obtained in 32% yield. However, it should be noted that shorter tryptophan containing dipeptide derivatives **8k** and **8 l** offered dimer macrocycles **9k** and **9 l** in 30% and 36% yields, respectively, suggesting that the peptide chain length control was particularly critical for C(7) cyclization (Fig. 5a). Notably, peptides containing amino acids with active functional groups, such as Phe (**8c** and **8e**), Glu (**8c**), Asp (**8d** and **8j**) and Arg (**8j**), had little impact on the reactions, thereby demonstrating the high reliability of our approach in assembling macrocyclic peptides with unique architectures.

Next, the traceless removal of *N*-pivaloyl group was performed with traditional methods[56–59]. However, no desired products were obtained under alkaline conditions. After condition optimization, we found that trifluoroacetic acid could successfully remove both the acid-sensitive protecting groups and the directing group *N*-pivaloyl group, and offered unprotected cyclic peptide at room temperature (Fig. 5b, Supplementary Figures 7 and 8). On the contrary, multiple steps are required to remove the directing and protecting groups of functionalized peptides obtained by Wang and Liu's method[40,43]. This indicates that the functionalized peptides obtained through our method could be further processesed conveniently. Furthermore, most of our cyclic peptides showed minimal cytotoxicity on the tested normal cells MIHA and LO2 (Supplementary Figure 14). Taken together, this late-stage peptide functionalization strategy is capable of producing macrocyclic peptides via regioselective C(7) Trp-maleimide crosslinkage.

## Potential thiol-ene click reaction applications

To further examine the robustness of this rhodium (III) catalytic system, gram scale C(7)-H maleimidation was carried out, and no significant loss of catalytic efficiency was observed (Supplementary Figure 9).

In general, maleimide has been widely utilized as an efficient functional group for click chemistry. Therefore, we investigated whether the maleimide-decorated tryptophan derivatives would still maintain good reactivity toward sulfuryl groups. Interestingly, the maleimide-decorated tryptophan could be easily reacted with sulfhydryl group of cysteine to obtain peptide **12a** in 4 min (Fig. 6). We further investigated whether these simply modified derivatives still possessed properties of click chemistry. Indeed, the maleimide-modified substrate **3a** still could react rapidly with sulfhydryl groups and gave the desired product of more than 80% within 5 min (Supplementary Figure 10). It is worth noting that, in comparison to the product of maleimide and cysteine obtained through Michael addition, which was not stable and would decompose over time[60], the product obtained from the trp-substituted maleimide **3a** and cysteine was more stable under physiological conditions (Supplementary Figure 10). The click conjugation of more C(7)-maleimide-modified tryptophan/peptides to biomolecules containing sulfhydryl groups were further investigated. As shown in Fig. 6, the maleimide-modified tryptophan **3n** was coupled with thiol-bearing amino acid to construct the dipeptide **12b** (93%) in only 4 min. Next, the maleimide modified peptides **5c** and **5e** were further coupled to amino acids by Michael addition to offer more complex peptides **12c** and **12d** (85% and 90%, respectively) within short reaction time. Notably, the complex peptide **7d** and hybrid molecule **7 f**, which were synthesized by C(7)-H

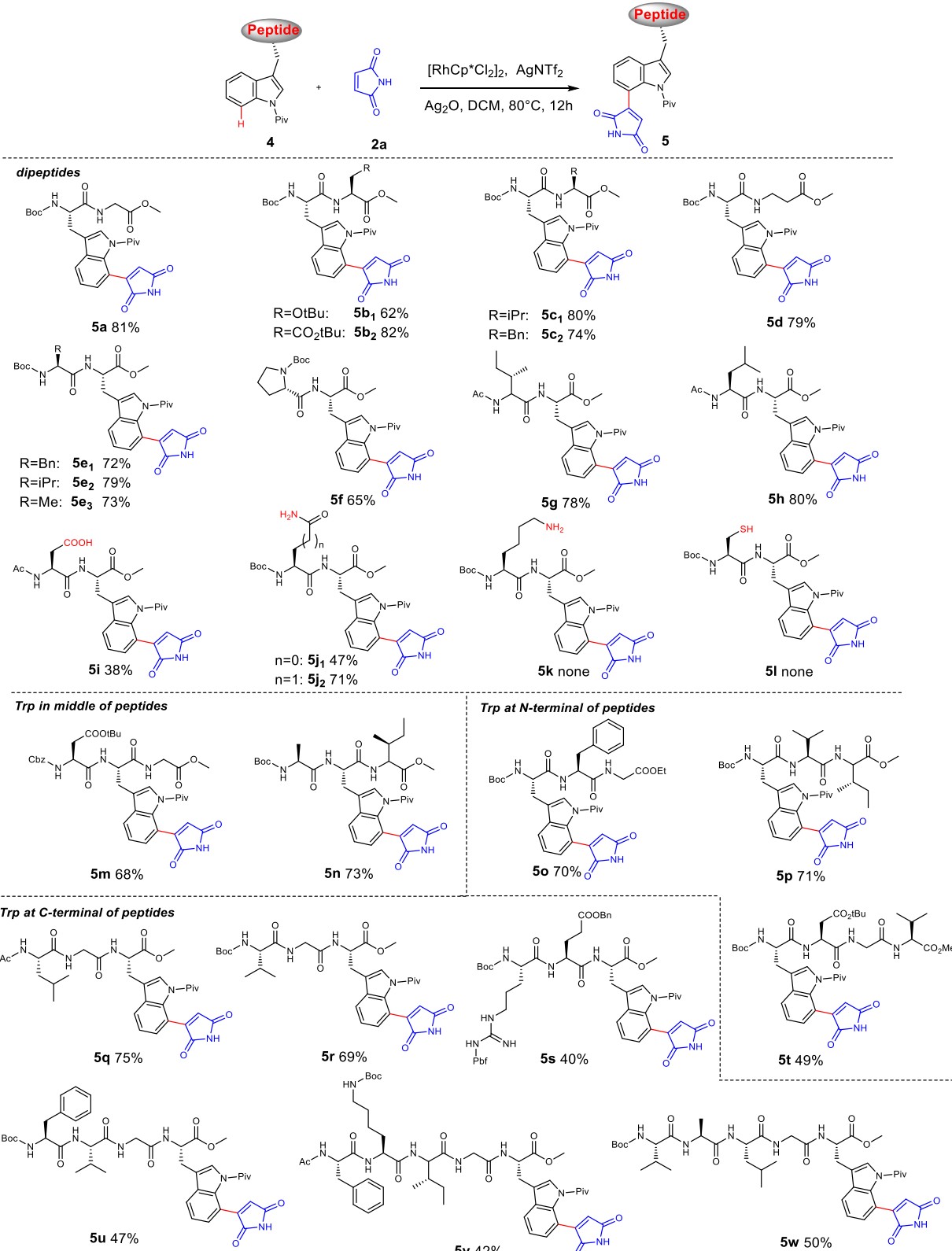

**Fig. 3 | Substrate scope for maleimidation of dipeptides and complex peptides.** Reaction conditions: **4** (0.2 mmol), **2a** (0.6 mmol), [RhCp*Cl$_2$]$_2$ (5 mol%), AgNTf$_2$ (20 mol%), Ag$_2$O (1.5 eq), DCM (1.5-2 mL), 80 °C, 12 h.

activation chemical ligation, could also be rapidly coupled to thiol-containing biomolecules to generate hexapeptide **12e** and hybrid molecule **12f**, which could realize the linkage of three functional molecules, with excellent yields (85% and 90%, respectively).

Moreover, the maleimide cyclic peptide **9d** could be rapidly coupled to the thiol-containing peptide, and the complex peptide **12g** was obtained in 87% yield, which realized further extension of the cyclic peptide. These data demonstrated that this class of peptides could

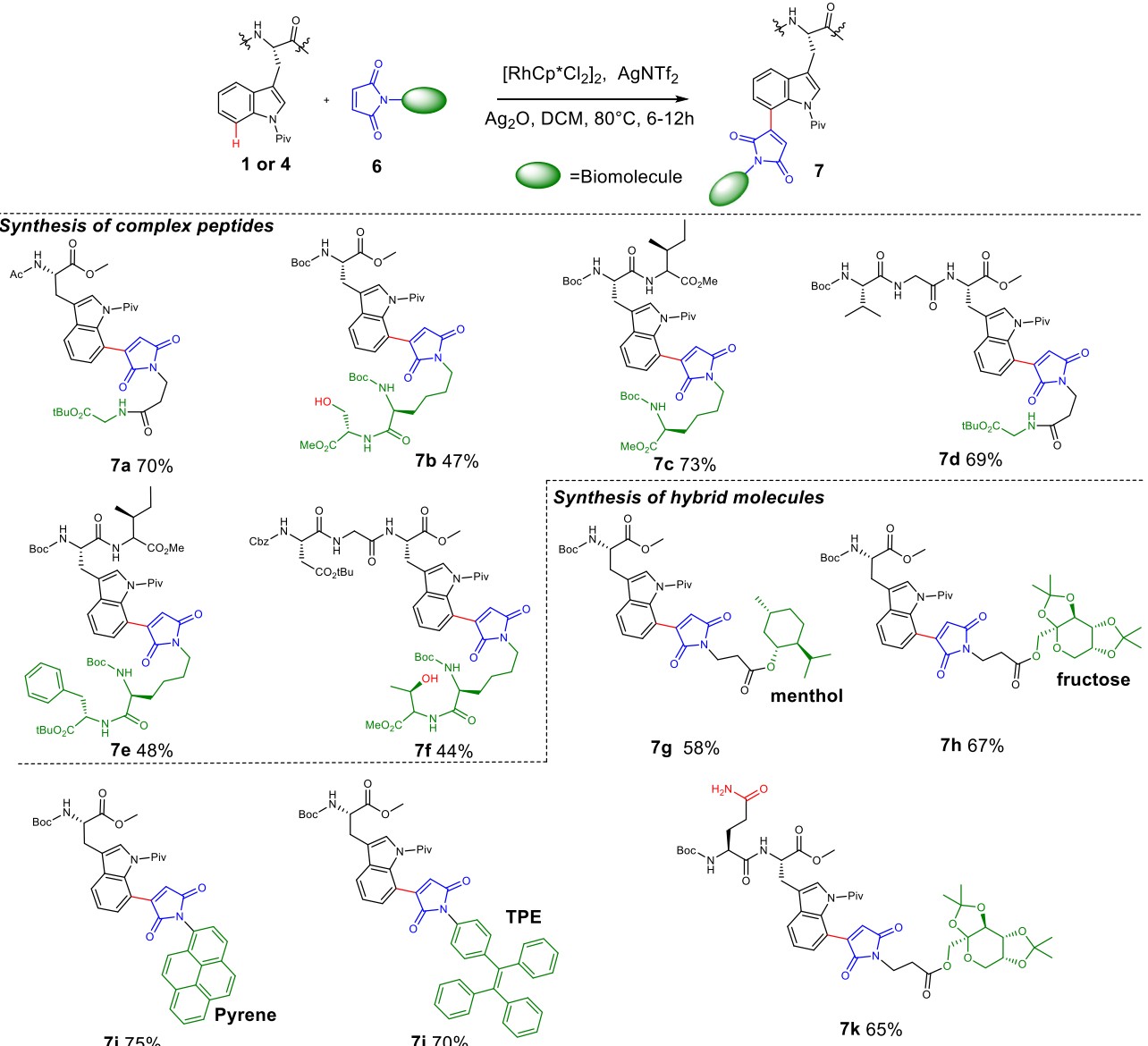

**Fig. 4 | Synthesis of complex peptides and hybrid molecules through C(7)-H activation chemical ligation.** Reaction conditions: **1** or **4** (0.2 mmol), **6** (0.6 mmol), [RhCp*Cl₂]₂ (5 mol%), AgNTf₂ (20 mol%), Ag₂O (1.5 eq), DCM (1.5 mL), 80 °C, 6–12 h.

easily label thiol-bearing molecules. In addition, these peptides are more versatile than the traditional maleimide derivatives which are widely used in ADC construction, indicating the potential of our approach for ADC/PDC drug development.

To evaluate the bioactivity of cyclic peptides, a cyclic peptide **10a** containing RGD sequence and a fluorescent probe **10b** with thiol groups were synthesized (Fig. 7a, Supplementary Figures 8 and 11), and the conjugated products of **10a** and **10b** were detected via liquid chromatography-mass spectrometry (LC-MS) (Supplementary Information page S44). In order to investigate the interaction between cyclic peptides and integrins, we conducted binding assays using surface plasmon resonance (SPR) technique. We selected integrin receptors, particularly $\alpha v\beta_3$, $\alpha v\beta_5$, and $\alpha_5\beta_1$, which are of clinical interest in anti-cancer therapy due to their expression on the surface of cancer cells and tumor neovasculature, as well as their role in mediating angiogenesis, tumor growth, and metastasis[61,62]. Initially, the integrins were immobilized on CM 5 chips using the amine coupling method. These chips were subsequently employed to measure the binding parameters of **10a** and the commercially available **10c** cyclo-

(RGDfK (FITC)) (analyte). As shown in Fig. 7b, the response unit (RU) values for the binding of the cyclic peptide **10a** to integrin $\alpha v\beta_3$ were concentration dependent, with the determined equilibrium dissociation constant $K_D$ of $6.75\times10^{-7}$ M being lower than the $K_D$ value of $6.55\times10^{-5}$ M for the control peptide **10c**. Supplementary Table 2 displays the $K_D$, $K_a$ and $K_d$ values of cyclic peptides **10a** and **10c** bound by different integrins in the SPR assay, indicating that the cyclic peptide **10a** exhibits a better binding affinity and selectivity than **10c**. Next, we utilized a flow cytometry assay with HUVEC cells as a reference to determine the number of $\alpha v\beta_3$ in HeLa and A549 cells. As shown in Fig. 7c, the number of $\alpha v\beta_3$ integrins in HeLa and A549 cells was 0.18 and 1.26-fold of that of HUVEC cells, respectively. Moreover, the binding affinity of **10a** to $\alpha v\beta_3$-overexpressed A549 cells was investigated, with integrin $\alpha v\beta_3$-lowexpressed HeLa cells serving as negative control and **10c** cyclo-(RGDfK (FITC)) used as a positive control. As shown in Fig. 7da or Fig. 7db, A549 cells incubated with **10a** and **10b** or incubated with **10c** exhibited robust fluorescence signals. Competitive experiments were next performed to assess the binding ability of **10a** to A549 cells (Fig. 7dc). We

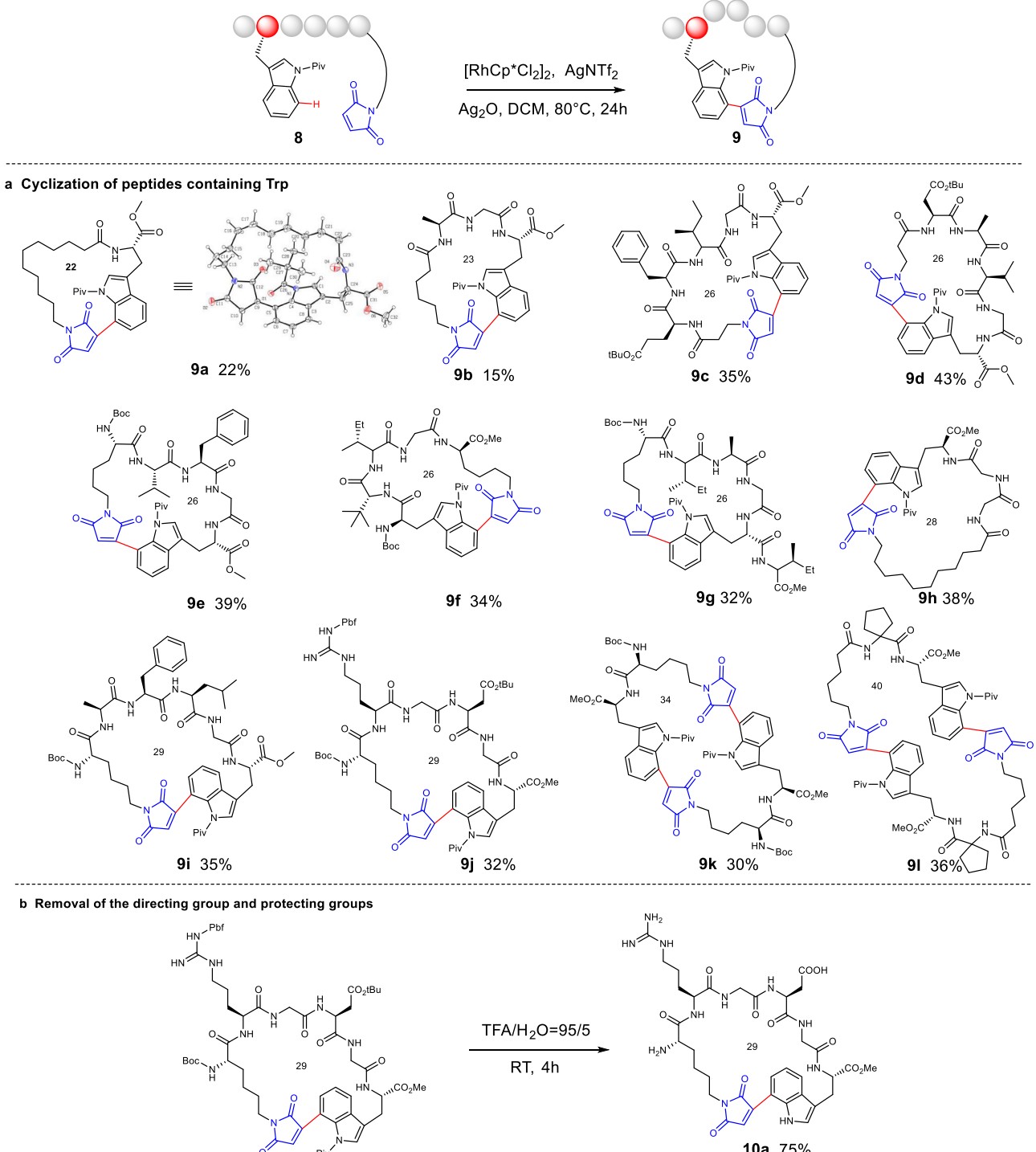

**Fig. 5 | Substrate scope for stapled peptides and removal of the directing group and protecting groups. a** Reaction conditions: **8** (0.12 mmol), [RhCp*Cl$_2$]$_2$ (10 mol%), AgNTf$_2$ (40 mol%), Ag$_2$O (1.5 eq), DCM (10 mL), 80 °C, 24 h; **b** Reaction conditions: **9j** (0.02 mmol), TFA/H$_2$O = 95/5 (1 mL, $v/v$), room temperature, 4 h.

observed fluorescence signals in both RhB and FITC channels. In contrast, HeLa cells subjected to the same treatments exhibited only weak fluorescence signals (Fig. 7dd). Fig. 7d shown the red fluorescence was stronger than the green emission based on fluorescence quantification analysis demonstrating that **10a** has a stronger binding affinity to integrin αvβ$_3$ than the commercially available cyclo-(RGDfK (FITC)) (**10c**). Taken together, the maleimide-decorated cyclic peptides may be potentially used as click functional groups for applications in chemical biology and medicinal chemistry.

Doxorubicin (DOX) is one of the most effective anticancer drugs and has been successfully used in clinical practice[63]. However, DOX cannot differentiate between cancer cells and normal cells, which may induce unwanted side effects and severe toxicity[64]. Compared with traditional small-molecule anticancer drugs, the peptide-drug conjugates (PDCs) have enhanced targeting specificity and water solubility[65]. Based on these advantages, to further explore the function of **10a**, we designed and prepared a anticancer PDC drug compound **RGD-GFLG-DOX** containing the tetrapeptide linker Gly-Phe-Leu-Gly,

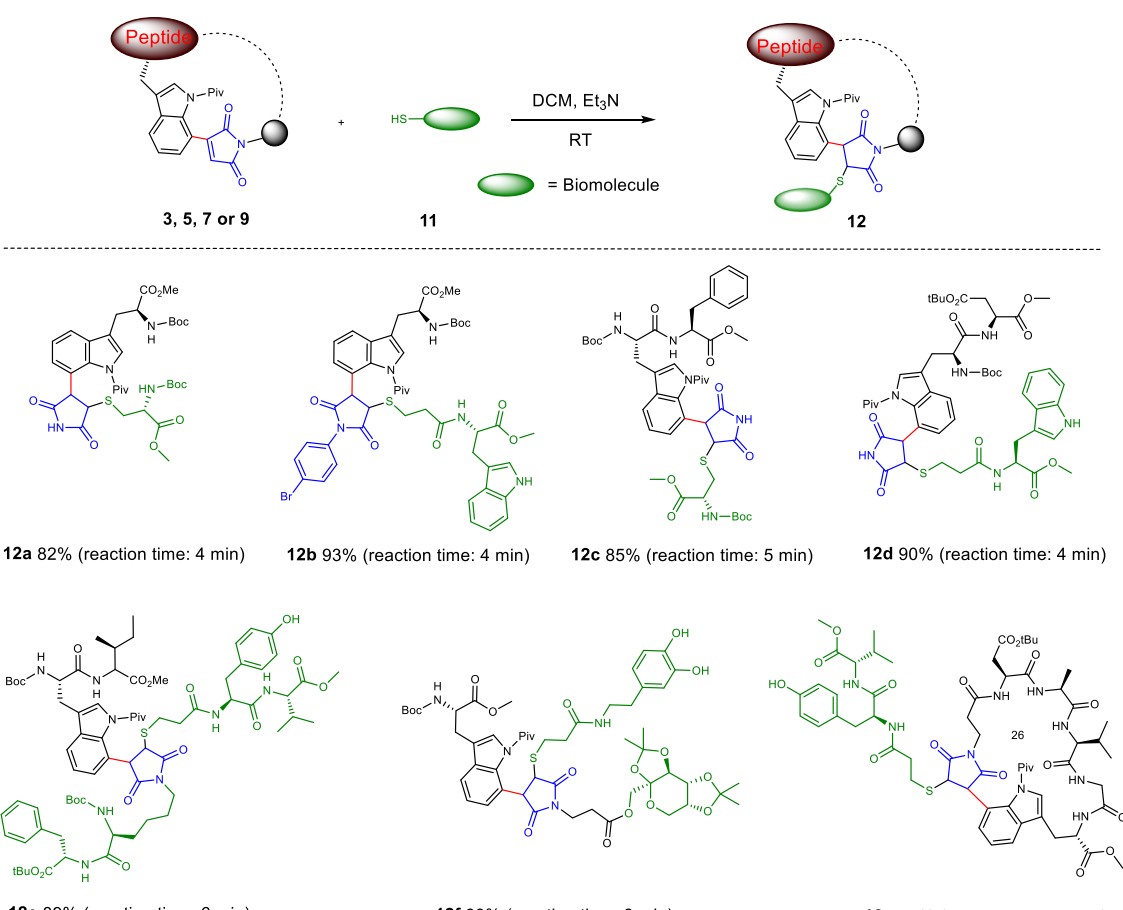

**Fig. 6 | Conjugation of C(7) maleimidated tryptophan/peptide with biomolecule containing sulfhydryl groups.** Reaction conditions: **3, 5, 7** or **9** (0.04 mmol), **11** (0.08 mmol), Et$_3$N (10 μL), DCM (1 mL), room temperature, 3–6 min.

which can be cleaved in presence of cathepsin B, a highly upregulated enzyme in malignant tumors[66], to release the drug (Fig. 8a, Supplementary Figure 12). **RGD-GFLG** was synthesized as a control (Fig. 8a, Supplementary Figure 13). The inhibitory effects of **RGD-GFLG-DOX** on cancer cell lines were assessed using cytotoxicity assay. Specifically, the effects of **RGD-GFLG-DOX** were evaluated on integrin αvβ$_3$-positive cancer cell lines, including A549 and U87MG cells, integrin αvβ$_3$-negative cancer cell lines such as HeLa and MCF-7 cells, as well as normal cell lines, namely LO2 and MIHA cells[67–69]. **RGD-GFLG-DOX** exhibited a lower cytotoxicity on HeLa, MCF-7, LO2 and MIHA cells, but a stronger cytotoxicity than DOX on A549 and U87MG cells (Fig. 8b). For comparison, **RGD-GFLG** demonstrated minimal cytotoxicity. In addition, the cytotoxicity of **RGD-GFLG-DOX** with various concentrations (0–40 μM) on A549 and U87MG cells was studied. The results showed that the cytotoxicity of **RGD-GFLG-DOX** on A549 and U87MG cells was dose-dependent. These indicate that **RGD-GFLG-DOX** has a good specificity and inhibitory activity toward integrin αvβ$_3$-overexpressed A549 and U87MG cells.

## Discussion

In conclusion, we developed a robust and regioselective rhodium-catalyzed methodology for C(7)-H Trp maleimidation. This reaction served as an efficient tool for peptide/drug modification, ligation, and particularly peptide cyclization, confirming its promising potential in pharmaceutical chemistry and drug synthesis. Notably, this catalytic system is not limited by the Trp position in the peptides. We also demonstrated that tryptophan-substituted maleimide could be used as an effective click functional group to rapidly react with sulfhydryl groups. Moreover, the introduced *N*-pivaloyl

directing group and protecting groups of the peptides could be removed in a single step, providing a more convenient approach compared to the previous methods[40,43], which require multi-step removal of the corresponding directing groups and peptide protection groups (Supplementary Figure 2). Additionally, cyclic peptide **10a** exhibited excellent binding affinity to integrin αvβ$_3$, indicating its good drug-like properties. With rational design, **RGD-GFLG-DOX**, which is a stapled PDC, displayed higher selectivity, stronger binding affinity and better cell penetrability than the more commonly used DOX. The proposed strategy for rapid preparation of stapled peptides is expected to further improve PDC formulation.

## Methods

### General

For reagents, equipments and detailed experimental procedures, see Supplementary Methods.

### General Maleimidylation of Trp/Trp containing peptides.

Typically, the Trp containing amino acid and peptide substrate (0.2 mmol), [RhCp*Cl$_2$]$_2$ (6.2 mg, 0.01 mmol) was suspended in 1.5-2 mL DCM, then maleimide/*N*-substituted maleimide derivatives (0.6 mmol), AgNTf$_2$ (15.5 mg, 0.04 mmol) and Ag$_2$O (69.5 mg, 0.3 mmol) were added. The tube was sealed, and the mixture was heated to 80 °C for 6–12 h. After cooling to ambient temperature, diluted with DCM and passed through a short celite pad, the solvent was evaporated in vacuum to get the crude product, which was then further purified by flash column or PTLC.

### General protocol for the peptides cyclization.

Typically, the linear peptide containing Trp (0.12 mmol), [RhCp*Cl$_2$]$_2$ (7.4 mg, 0.012 mmol)

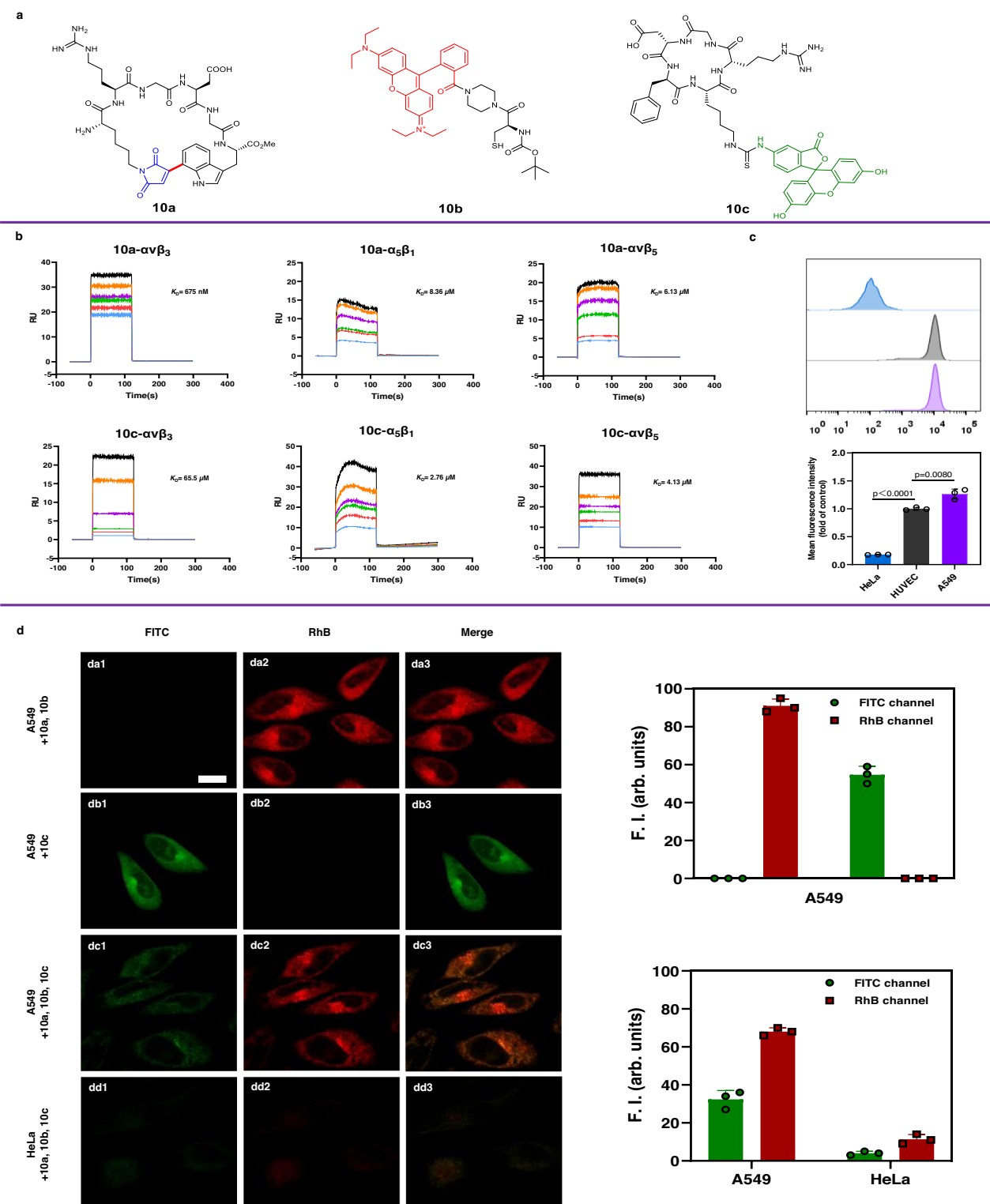

**Fig. 7 | Study on the affinity of cyclic RGD peptides to integrins. a** The structures of cyclic RGD peptides **10a** and **10c** and fluorescent probes **10b** with thiol groups. **b** SPR sensorgrams characterizing the affinity of peptides **10a** and **10c** to integrins targets. **c** Flow cytometry assay determining the amount of integrins in HeLa and A549 cells. Living cells were pretreated with 5 μM cyclic peptides **10a** for 60 min, and then with 5 μM **10b** for 15 min. HUVEC cells were served as a reference. Data were presented as mean ± S.D. $n = 3$ biologically independent samples per group. *P* values obtained from two-tailed unpaired *t*-test. **d** Confocal fluorescence images of A549 or HeLa cells pretreated with 5 μM cyclic peptides **10a** d**a**), **10c** d**b**) or **10a** and **10c** d**c**) and **dd**) for 60 min, and then with 5 μM **10b** d**a**), d**c**) and **dd**) for 15 min, upon one-photon excitation at 488 nm (**10c**) and 552 nm (**10a** + **10b**). Scale bar: 20 μM. The images were collected at 500–550 nm (**10c**) and 580–650 nm (**10a** + **10b**). Fluorescence quantitative analysis of FITC and RhB channels in A549 and HeLa cells. Data are expressed as mean ± SD (experiment times $n = 3$).

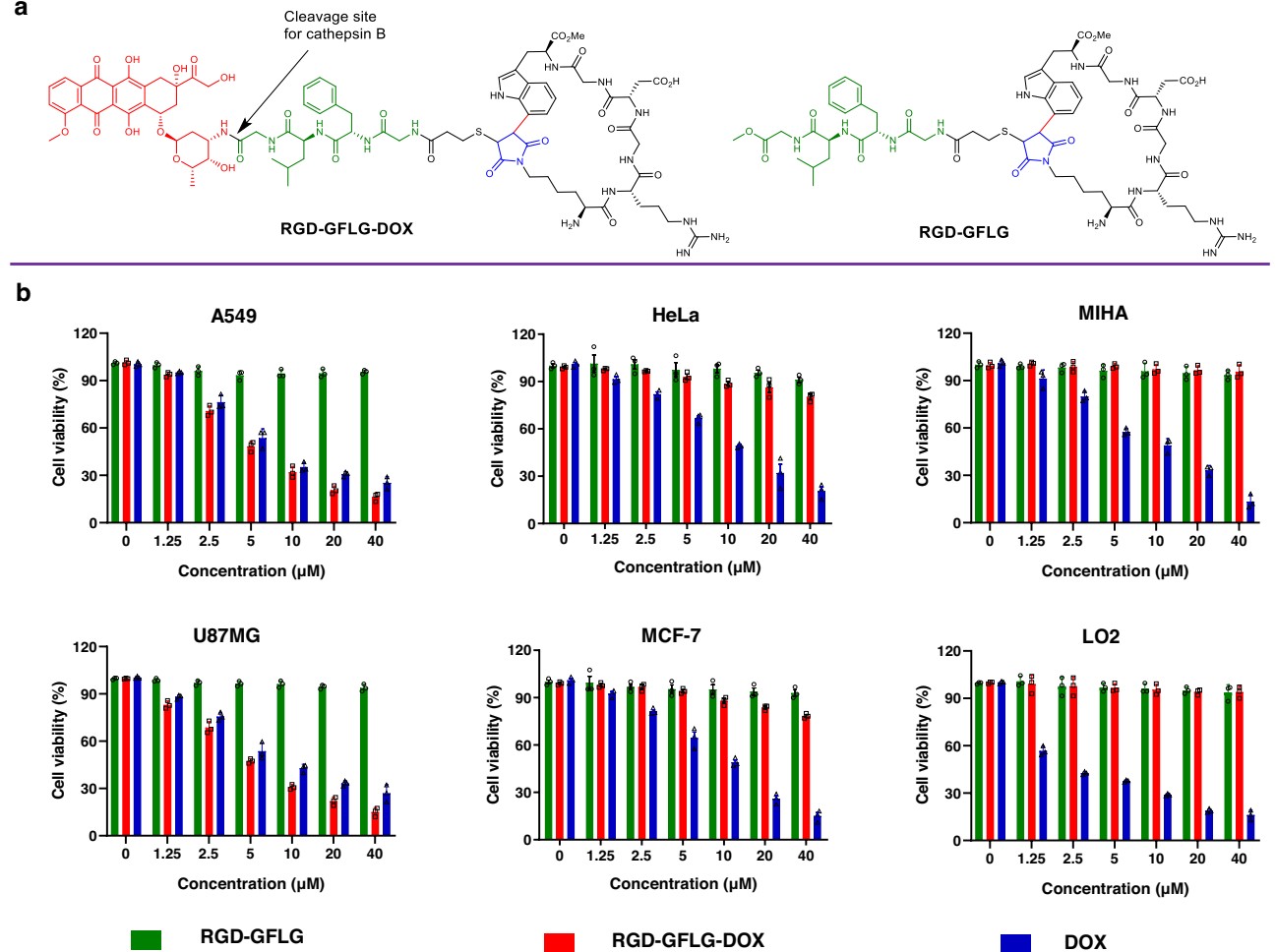

**Fig. 8 | Cytotoxicity evaluation of RGD-GFLG-DOX/RGD-GFLG/DOX in vitro.**
**a** The structure of **RGD-GFLG-DOX/RGD-GFLG**. **b**) Cytotoxicity of **RGD-GFLG-DOX**, **RGD-GFLG** and DOX on A549, U87MG, HeLa, MCF-7, MIHA and LO2 cells. Cells were incubated with various concentrations of **RGD-GFLG-DOX, RGD-GFLG** and DOX (0–40 µM). After adding drugs, the cells were further incubated for another 36 h. Data were presented as mean ± S.D. $n = 3$ biologically independent samples per group.

was suspended in 10 mL DCM, then AgNTf$_2$ (18.6 mg, 0.048 mmol) and Ag$_2$O (41.7 mg, 0.18 mmol) were added. The tube was sealed, and the mixture was heated to 80 °C for 24 h. After cooling to ambient temperature, the mixture was diluted with DCM and passed through a short celite pad. The solvent was evaporated in vacuum to get the crude product, which was further purified by flash column or PTLC.

Cell culture. Following cell lines were used in this study: A549 (TCH-C116), HeLa (TCH-C193), MCF-7 (TCH-C247), U87MG(TCH-C367), MIHA (CL0469), LO2 (CL0192), HUVEC (TCH-C406), And A549 (TCH-C116), HeLa (TCH-C193), MCF-7 (TCH-C247), U87MG(TCH-C367) and HUVEC (TCH-C406) originally purchased from Suzhou Starfish Biotechnology Co. LTD. MIHA (CL0469) and LO2 (CL0192) originally purchased from Hunan Fenghui Biotechnology Co. LTD.

MIHA and LO2 cells were cultured in RPMI 1640 (Hyclone) medium supplemented with 10% FBS (Titan). A549, HeLa, MCF-7, U87MG and Huvec cells were cultured in DMEM (Hyclone) medium supplemented with 10% FBS. All cells were cultured at 37 °C in an atmosphere of 5% CO$_2$. All media contained 100 units/mL penicillin and 100 µg/mL streptomycin.

**Confocal fluorescence imaging of peptides in A549 and HeLa cells.**
A549 and HeLa cells were cultured in DMEM high glucose media supplemented with 10% fetal bovine serum, 1% Penstrep, 0.2% Amphotericin B. The cells were grown overnight at 37 °C incubator with 5% CO$_2$. A549 and HeLa cells were seeded at a density of 3×10$^5$ cells in 35 mm glass-bottomed dish and kept overnight prior to cell imaging studies. After 24 h, the cells were washed twice with warm DMEM media, incubated at 37 °C with 5 µM peptides for 60 min, and then pretreated with 5 µM probe for 15 min. Images were taken using the Zeiss LSM 800 confocal fluorescence microscope.

**Flow cytometry analysis for cell surface integrin proteins.** Cells were cultured in DMEM high glucose medium supplemented with 10% fetal bovine serum, 1% Streptococcus vale, 0.2% amphotericin B. Cells are grown overnight with 5% CO$_2$ in a 37 °C incubator. Cells are seeded in 6-well plate dishes at a density of 3×10$^5$ cells and kept overnight before cell detection. After 24 h, wash the cells twice with warm DMEM medium, incubate with 5 µM peptide for 60 min at 37 °C, and then pretreat with a 5 µM probe for 15 min. Before measurement by flow cytometry, cells were scraped off gently and collected into a clean 2 mL centrifuge tube. Then, cells were spun down (1000 rpm, room temperature, 3 min). After discarding the supernatant, 1 mL of warm PBS was added gently to resuspend the cell pellet. Finally, cells were analyzed on a Guava (Millipore) flow cytometer equipped with a 488 nm Ar laser, and fluorescence was collected by PE channel. And the flow cytometric data were analyzed with flow analysis software Flow Jo.

**Cytotoxicity was assessed by MTT assay.** Various cells were cultured in DMEM medium in 96-well microplates in at 37 °C under 5% CO$_2$ for 12 h. The medium was next replaced with fresh medium containing

various concentrations of cyclic peptides, **RGD-GFLG, RGD-GFLG-DOX** and DOX (0–40 μM). Each concentration was tested in triplicate. After 36 h, cells were washed twice with PBS buffer and incubated with 0.5 mg/mL MTT reagent for 4 h at 37 °C. 150 μL DMSO was then added to dissolve formazan. Measure the absorbance at 510 nm in a microplate reader. Cell viability (%) was calculated according to the following equation: Viability = (mean Abs. of treated wells/mean Abs. of control wells) × 100%.

## Reporting summary
Further information on research design is available in the Nature Portfolio Reporting Summary linked to this article.

## Data availability
The authors declare that the data supporting the findings of this study are available within the paper and its Supplementary Information files. The data generated in this study are provided in the Source Data file. Source data are provided with this paper.

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

## Acknowledgements

The authors are grateful to the National Natural Science Foundation of China (Nos. 22177103, 21472172 to Q. Z.).

## Author contributions

Conceptualization, P.W. and J.L.; Methodology, P.W.; Investigation, P.W., X.Z., Z.Y., J.Y., J.J., M.F. and J.L.; Writing—Original Draft, P.W.; Writing—Review & Editing, P.W., J.L., X.Z., K., J.G. and Q.Z.; Funding Acquisition, Q.Z.; Resources, Q.Z. and J.G.; Supervision, Q.Z. and Y.Z.

## Competing interests

The authors declare no competing interests.
