## [Peer Review File · Nature Communications]

REVIEWER COMMENTS

Reviewer #1 (Remarks to the Author):

This manuscript described the development of a Rh-catalyzed C-H maleimidation method for the late-stage modification and macrocyclization of oligopeptides. With the assistance of a directing group pre-installed at the N(Trp), the modification occurs specifically at the C7 of Trp residues. The authors demonstrated the applicability of this method by synthesizing a variety of amino acid derivatives, short peptide derivatives and cyclic peptides. The bioactivity of the resulting cyclic peptides was demonstrated by a cyclic RGD peptide derivative generated via this method. Overall, this study provides a valuable chemical method for modification of peptides at the C7 of Trp residues. This reviewer would suggest the publication of this manuscript with several comments:

1. Is this method compatible with unprotected amino acids, especially Lys, Asn, Gln, Cys, and Ser? This is a common concern for almost all peptide C-H modification methods.
2. In the Figure 7B, the authors carried out a competitive experiment of 10a with the commercially available 10c cyclo-(RGDFK (FITC)). To better understanding of the binding affinity of 10a to integrin $\alpha\beta3$ overexpressed A549 cells, the authors should carry out the Confocal fluorescence images of A549 and HeLa cells treated with 10a and 10c, respectively, for comparison.
3. In the last paragraph of Result part, the authors designed a PDC drug, RGD-GFLG-DOX, and exhibited good specificity and inhibitory activity toward integrin $\alpha\beta3$ -overexpressed A549 cells. However, to demonstrate the selective inhibitory effect to cancer cells of PDC drug, it necessary to investigate the inhibitory effect of RGD-GFLG-DOX toward normal cells as control.

Reviewer #2 (Remarks to the Author):

The manuscript by describes Rh catalyzed C(7)-maleimidation of tryptophan containing peptides. Although the scope of the protocol is impressive, especially the compatibility of the methodology in presence of various peptides is excellent. Additionally, tryptophan-substituted maleimide have been employed as a novel click functional group to rapidly react with sulfhydryl groups. However, the work lacks novelty to get published in Nature Communication. In recent years there are numerous reports on C(sp²)-H functionalization of tryptophan peptides including in C2, C4, and C7 sites. This is another methodology to insert maleimide fragments to tryptophan. In absence of clear novelty, I cannot recommend for publication in Nat. Chem.

Reviewer #3 (Remarks to the Author):

The manuscript reports a synthetic method for modifying tryptophan with a maleimide group. This idea enables a subsequent conjugation to the maleimide using thiolate. The reaction procedure is of interest for peptide bioconjugation and particularly for cyclization of peptide.

The following points should be addressed:

1. Figure 1 : Please check for typo. Excellent Chemical
2. Figure 7: Line 208, page 15. The binding of 10a was investigated. If so, please provide the K_a (of K_d). It will be good idea to give the K_a (K_d) of the control cyclic RGD peptide.
3. RGD peptides, including cyclic peptides bind to many integrins. Since the cycle is different than the control cyclicRGD, it will be an excellent idea to verify that 10a has similar selectivity than the control for different integrins
4. HUVECs are the cells that are generally used for $\alpha\beta3$ testing. Please measure/give the

number of alphavbeta3 on the A549 cells.

5. The manuscript is about a new chemical methodology. Discussion should mention advantages and disadvantages of this methodology versus other ones. What about the compatibility with peptides containing cysteines? How to remove the reactants? How toxic they are?

RESPONSES TO REVIEWER COMMENTS

Reviewer #1

This manuscript described the development of a Rh-catalyzed C-H maleimidation method for the late-stage modification and macrocyclization of oligopeptides. With the assistance of a directing group pre-installed at the N(Trp), the modification occurs specifically at the C7 of Trp residues. The authors demonstrated the applicability of this method by synthesizing a variety of amino acid derivatives, short peptide derivatives and cyclic peptides. The bioactivity of the resulting cyclic peptides was demonstrated by a cyclic RGD peptide derivative generated via this method. Overall, this study provides a valuable chemical method for modification of peptides at the C7 of Trp residues. This reviewer would suggest the publication of this manuscript with several comments:

Q1. Is this method compatible with unprotected amino acids, especially Lys, Asn, Gln, Cys, and Ser? This is a common concern for almost all peptide C-H modification methods.

Response:

We thank the reviewer for considering whether this method is compatible with unprotected amino acids. As shown in **Figures 3 and 4**, the different peptides containing unprotected active amino acid side chains were synthesized to verify the compatibility of this method with unprotected amino acids. The results demonstrated that the method is compatible with some unprotected active amino acids, including Asp (**5i**), Asn (**5j1**), Gln (**5j2** and **7k**), Ser (**7b**) and Thr (**7f**). Meanwhile, amino acids, such as Lys (**5k**) and Cys (**5l**) could not give the desired products. Specifically, the amino group on the side chain of Lys underwent Michael addition with maleimide, resulting in the addition product **5k'**. The -SH of Cys is susceptible to oxidation which render it incompatible with the reaction. The NMR and HRMS data of all these compounds were included in SI. Results and discussion are added on page 8 and 11 highlighted in yellow.

Figure 3. Substrate scope for maleimidation of dipeptides and complex peptides. Reaction conditions: **4** (0.2 mmol), **2a** (0.6 mmol), [RhCp*Cl₂]₂ (5 mol%), AgNTf₂ (20 mol%), Ag₂O (1.5 eq), DCM (1.5-2 mL), 80°C, 12 h.

Figure 4. Synthesis of complex peptides and hybrid molecules through C(7)-H activation chemical ligation. Reaction conditions: **1** or **4** (0.2 mmol), **6** (0.6 mmol), [RhCp*Cl₂]₂ (5 mol%), AgNTf₂ (20 mol%), Ag₂O (1.5 eq), DCM (1.5 mL), 80°C, 6-12 h.

Q2. In the Figure 7B, the authors carried out a competitive experiment of 10a with the commercially available 10c cyclo-(RGDfK (FITC)). To better understanding of the binding affinity of 10a to integrin $\alpha\beta3$ overexpressed A549 cells, the authors should carry out the Confocal fluorescence images of A549 and HeLa cells treated with 10a and 10c, respectively, for comparison.

Response:

We thank the reviewer for the valuable suggestion! We supplemented the experiment by comparing confocal fluorescence images of A549 cells and HeLa cells treated with **10a** and **10c**. As shown in **Figure 7Da** or **Figure 7Db**, A549 cells incubated with **10a** and **10b** or incubated with **10c** exhibited robust fluorescence signals. To further assess the binding ability of **10a** to A549 cells, competitive experiments were performed by using **10c** as control. As shown in **Figure 7Dc**, fluorescence signals were observed in both RhB and FITC channels. In contrast, HeLa cells subjected to the same treatments exhibited only weak fluorescence signals (**Figure 7Dd**). Notably, the red fluorescence was stronger than the green emission based on fluorescence quantification analysis (**Figure 7D**). Taken together, these results demonstrated that **10a** displays stronger

binding affinity to integrin $\alpha\beta_3$ than the commercially available cyclo-(RGDfK (FITC)) (10c). Results and discussion are added on page 17-18 highlighted in yellow.

Figure 7. A) The structures of cyclic RGD peptides **10a** and **10c** and fluorescent probes **10b** with thiol groups. D) Confocal fluorescence images of A549 or HeLa cells pretreated with 5 μ M cyclic peptides **10a** a), **10c** b) or **10a** and **10c** c) and d) for 60 min, and then with 5 μ M **10b** a), c) and d) for 15 min, upon one-photon excitation at 488 nm (**10c**) and 552 nm (**10a+10b**). Scale bar: 20 μ M. The images were collected at 500-550 nm (**10c**) and 580-650 nm (**10a+10b**). Fluorescence quantitative analysis of FITC channel and RhB channel in A549 and HeLa cells. Data are expressed as mean \pm SD (experiment times n = 3).

Q3. In the last paragraph of Result part, the authors designed a PDC drug, RGD-GFLG-DOX, and exhibited good specificity and inhibitory activity toward integrin $\alpha\beta_3$ -overexpressed A549 cells. However, to demonstrate the selective inhibitory effect to cancer cells of PDC drug, it necessary to investigate the inhibitory effect of RGD-GFLG-DOX toward normal cells as control.

Response:

We thank the reviewer for raising the question regarding the inhibitory effect of **RGD-GFLG-DOX** on normal cells, which is essential to highlight the strength of our study. We added integrin $\alpha\beta_3$ -positive cancer line U87MG cells, integrin $\alpha\beta_3$ -negative cancer line MCF-7 cells, normal cell lines LO2 and MIHA cells.¹⁻³ We also synthesized **RGD-GFLG** compounds without DOX as controls (**Figure 8A** and **Scheme S9**, its characterization data was shown in Supplementary Information, page S44). The inhibitory effects of **RGD-GFLG-DOX** on cancer cell lines were assessed. **RGD-**

GFLG-DOX exhibited a lower cytotoxicity to integrin $\alpha\beta_3$ -negative cell lines, i.e., HeLa and MCF-7 cells and the normal cell lines, i.e., LO2 and MIHA cells, but a stronger cytotoxicity than DOX to A549 and U87MG cells (**Figure 8B**). For comparison, **RGD-GFLG** demonstrated to be much lower cell cytotoxicity. In addition, the cytotoxicity of **RGD-GFLG-DOX** with various concentrations (0-40 μM) on A549 and U87MG cells was studied. The results showed that the cytotoxicity of **RGD-GFLG-DOX** on A549 and U87MG cells was dose-dependent. These indicate that **RGD-GFLG-DOX** has a good specificity and inhibitory activity towards integrin $\alpha\beta_3$ -overexpressed A549 and U87MG cells. These results and discussion were displayed on page 19 highlighted in yellow.

Figure 8. A) The structure of **RGD-GFLG-DOX/**RGD-GFLG. Cytotoxicity evaluation of **RGD-GFLG-DOX/** RGD-GFLG/DOX *in vitro*. B) Cytotoxicity of **RGD-GFLG-DOX**, **RGD-GFLG** and DOX on A549, U87MG, HeLa, MCF-7, MIHA and LO2 cells. Different cells were incubated with various concentrations of **RGD-GFLG-DOX**, **RGD-GFLG** and DOX (0-40 μM). After adding drugs, the cells were further incubated for another 36h. Data are given as mean \pm sd (n = 3).

References

1. Chatzisideri, T. et al. Integrin-Mediated Targeted Cancer Therapy Using c(RGDyK)-Based Conjugates of Gemcitabine. *J. Med. Chem.* **65**, 271-284 (2022).
2. Zhang, L. et al. Novel Integrin $\alpha\beta_3$ -Specific Ligand for the Sensitive Diagnosis of Glioblastoma. *Mol. Pharm.* **16**, 3977-3984 (2019).
3. Zhang, X. et al. Quantitative Analysis of Multiple Proteins of Different Invasive Tumor Cell Lines at the Same Single-Cell Level. *Small* **14**, e1703684 (2018).

Reviewer #2:

The manuscript by describes Rh catalyzed C(7)-maleimidation of tryptophan containing peptides. Although the scope of the protocol is impressive, especially the compatibility of the methodology in presence of various peptides is excellent. Additionally, tryptophan-substituted maleimide have been employed as a novel click functional group to rapidly react with sulfhydryl groups. However, the work lacks novelty to get published in Nature Communication. In recent years there are numerous reports on C(sp²)-H functionalization of tryptophan peptides including in C2, C4, and C7 sites. This is another methodology to insert maleimide fragments to tryptophan. In absence of clear novelty, I cannot recommend for publication in Nat. Chem.

Response:

We thank the reviewer for the comments and acknowledge the concern raised. We would like to highlight that some contributions of C-H functionalization on tryptophan have been made during the past decade by Lavilla/Albericio,^{1, 2} Ackermann,³⁻⁷ and Wang,⁸ among others.⁹⁻¹⁵ For example, Liu et al.¹³ successfully introduced maleimide into tryptophan and tryptophan-containing peptides at C(2) position by utilizing **N-pyridine** as directing group. Notably, most of them focused on the C(2) site owing to its high electron density, resulting in difficult to selectively C-H functionalize on the more inert C(4) and C(7) sites. Up to now, only limited examples of C(4)-H activation of tryptophan were achieved successfully assisted by directing groups.^{8, 16-18}

For Trp C(7) functionalization, Ackermann et al.¹⁹ recently reported a direct late-stage peptide C-H amidation at the tryptophan C(7) position by taking the N-pyridinyl group as directing group. However, they did not give the cyclization examples in this literature. Normally, to construct macrocyclic peptides, prefunctionalization, such as I, OTf, BR₂ and SnR₃, at C(7) position is essential.²⁰⁻²² To realize direct and site-selective cyclization via C(7)-H activation, we developed a Rh-catalyzed C(7) maleimidation and submitted the manuscript to Nat. Commun. on April 2022. Nearly at the same time, Wang and Shi²³ reported the Rh-catalyzed late-stage peptide ligation and macrocyclization of C(7)-H alkylation of Trp with alkenes, utilizing the **N-P^tBu₂** directing group on May 2022. However, these reactions are different. The maleimidated peptides reported in this manuscript maintains the double bond. This is a biorthogonal group, and could rapidly couple with thiol containing biomolecules, which have potentially wide applications in chemical biology and drug discovery. Wang's cyclic peptides were achieved by addition reactions, which lost the possibility for further modifications. Thus, by taking advantages of this methodology, a peptide drug conjugation (PDC) **RGD-GFLG-DOX** by click chemical reaction between maleimide-cyclopeptide **10a** and thiol compound **15** was successfully constructed, which exhibited excellent specificity for tumor cells.

Additionally, the removal of these directing and protecting groups requires multiple and time-consuming steps (**Figure S2A, B**). To address this issue, we introduced N-pivaloyl as directing group for maleimidation on C(7)-H tryptophan and further investigated the strategy for removing the directing groups. Although no desired product was observed under traditional deprotecting conditions,²⁴⁻²⁶ we successfully found that trifluoroacetic acid could remove the directing group and protecting group of the peptide at an ambient condition after extensive screening the conditions, offering the unprotected maleimide cyclic peptide (**Figure S2C**). We believe this finding will facilitate the further application of our method in the field of chemical biology, especially the construction of PDC.

Figure S2. General Procedure for Removal of Directing and Protecting Groups in Functionalized Peptides

Taken together, in contrast to all the highlighted previous works, we have in this study developed a novel, robust and regioselective rhodium-catalyzed methodology for late-stage diversification of Trp/Trp-containing peptides at the C(7) position, which exhibited significant application value in the fields of biochemistry, medicinal chemistry, and proteomics research.

We have added these comments on page 4 and 13 highlighted in yellow.

References

1. Mendive-Tapia, L. et al. Constrained Cyclopeptides: Biaryl Formation through Pd-Catalyzed C-H Activation in Peptides-Structural Control of the Cyclization vs. Cyclodimerization Outcome. *Chem. – Eur. J.* **22**, 13114-13119 (2016).
2. Mendive-Tapia, L. et al. New peptide architectures through C-H activation stapling between tryptophan-phenylalanine/tyrosine residues. *Nat. Commun.* **6**, 7160 (2015).
3. Kaplaneris, N. et al. Late-stage stitching enabled by manganese-catalyzed C-H activation: Peptide ligation and access to cyclopeptides. *Sci. Adv.* **7**, eabe6202 (2021).

4. Kaplaneris, N. et al. Chemodivergent manganese-catalyzed C-H activation: modular synthesis of fluorogenic probes. *Nat. Commun.* **12**, 3389 (2021).
5. Lorion, M.M., Kaplaneris, N., Son, J., Kuniyil, R. & Ackermann, L. Late-Stage Peptide Diversification through Cobalt-Catalyzed C-H Activation: Sequential Multicatalysis for Stapled Peptides. *Angew. Chem. Int. Ed.* **58**, 1684-1688 (2019).
6. Ruan, Z., Sauermann, N., Manoni, E. & Ackermann, L. Manganese-Catalyzed C-H Alkynylation: Expedient Peptide Synthesis and Modification. *Angew. Chem. Int. Ed.* **56**, 3172-3176 (2017).
7. Schischko, A. et al. Late-stage peptide C-H alkylation for bioorthogonal C-H activation featuring solid phase peptide synthesis. *Nat. Commun.* **10**, 3553 (2019).
8. Bai, Z., Cai, C., Sheng, W., Ren, Y. & Wang, H. Late-Stage Peptide Macrocyclization by Palladium-Catalyzed Site-Selective C-H Olefination of Tryptophan. *Angew. Chem. Int. Ed.* **59**, 14686-14692 (2020).
9. Dong, H., Limberakis, C., Liras, S., Price, D. & James, K. Peptidic macrocyclization via palladium-catalyzed chemoselective indole C-2 arylation. *Chem. Commun.* **48**, 11644-11646 (2012).
10. Kee, C.W. et al. 18F-Trifluoromethanesulfinate Enables Direct C-H 18F-Trifluoromethylation of Native Aromatic Residues in Peptides. *J. Am. Chem. Soc.* **142**, 1180-1185 (2020).
11. Liu, J. et al. A Peptide Stapling Strategy with Built-In Fluorescence by Direct Late-Stage C(sp²)-H Olefination of Tryptophan. *Chem. – Eur. J.* **26**, 16122-16128 (2020).
12. Liu, J., Wang, P., Zeng, W., Lu, Q. & Zhu, Q. Late-stage construction of stapled peptides through Fujiwara-Moritani reaction between tryptophan and olefins. *Chem. Commun. (Camb)* **57**, 11661-11664 (2021).
13. Peng, J., Li, C., Khamrakulov, M., Wang, J. & Liu, H. Rhodium(III)-Catalyzed C-H Alkenylation: Access to Maleimide-Decorated Tryptophan and Tryptophan-Containing Peptides. *Org. Lett.* **22**, 1535-1541 (2020).
14. Reay, A.J. et al. Mild and Regioselective Pd(OAc)₂-Catalyzed C-H Arylation of Tryptophans by [ArN₂]X, Promoted by Tosic Acid. *ACS Catal.* **7**, 5174-5179 (2017).
15. Terrey, M.J., Holmes, A., Perry, C.C. & Cross, W.B. C-H Olefination of Tryptophan Residues in Peptides: Control of Residue Selectivity and Peptide-Amino Acid Cross-linking. *Org. Lett.* **21**, 7902-7907 (2019).
16. Li, J.J., Mei, T.S. & Yu, J.Q. Synthesis of indolines and tetrahydroisoquinolines from arylethylamines by Pd(II)-catalyzed C-H activation reactions. *Angew. Chem. Int. Ed.* **47**, 6452-6455 (2008).
17. Liu, Q., Li, Q., Ma, Y. & Jia, Y. Direct Olefination at the C-4 Position of Tryptophan via C-H Activation: Application to Biomimetic Synthesis of Clavicipitic Acid. *Org. Lett.* **15**, 4528-4531 (2013).
18. Zhang, Q. et al. Direct C4-Acetoxylation of Tryptophan and Tryptophan-Containing Peptides via Palladium(II)-Catalyzed C-H Activation. *Org. Lett.* **23**, 4699-4704 (2021).
19. Wang, W. et al. Peptide Late-Stage Diversifications by Rhodium-Catalyzed Tryptophan C7 Amidation. *Chem* **6**, 3428-3439 (2020).
20. Jia, Y., Bois-Choussy, M. & Zhu, J. Synthesis of Diastereomers of Complestatin and Chloropeptin I: Substrate-Dependent Atropstereoselectivity of the Intramolecular Suzuki-Miyaura Reaction. *Angew. Chem. Int. Ed.* **47**, 4167-

- 4172 (2008).
21. Isley, N.A. et al. Total Synthesis and Stereochemical Assignment of Streptide. *J. Am. Chem. Soc.* **141**, 17361-17369 (2019).
 22. Deng, H. et al. Total Synthesis of Anti-HIV Agent Chloropeptin I. *J. Am. Chem. Soc.* **125**, 9032-9034 (2003).
 23. Liu, L. et al. P(III) -Directed Late-Stage Ligation and Macrocyclization of Peptides with Olefins by Rhodium Catalysis. *Angew. Chem. Int. Ed.* **61**, e202206177 (2022).
 24. Ruiz, M., Sánchez, J.D., López-Alvarado, P. & Menéndez, J.C. A systematic study of two complementary protocols allowing the general, mild and efficient deprotection of N-pivaloylindoles. *Tetrahedron* **68**, 705-710 (2012).
 25. Choi, I., Messinis, A.M. & Ackermann, L. C7-Indole Amidations and Alkenylations by Ruthenium(II) Catalysis. *Angew. Chem. Int. Ed.* **59**, 12534-12540 (2020).
 26. Behloul, C., Chouti, A., Guijarro, D., Nájera, C. & Yus, M. Reductive Removal of the Pivaloyl Protecting Group from Tetrazoles by a Naphthalene-Catalyzed Lithiation Process. *Synthesis* **47**, 507-510 (2014).

Reviewer #3:

The manuscript reports a synthetic method for modifying tryptophan with a maleimide group. This idea enables a subsequent conjugation to the maleimide using thiolate. The reaction procedure is of interest for peptide bioconjugation and particularly for cyclization of peptide.

The following points should be addressed:

Q1. Figure 1 : Please check for typo. Excellent Chemical

Response:

We thank the reviewer for pointing out the spelling error in **Figure 1**. We have corrected this in the revised manuscript. In addition, we have reviewed the entire manuscript to address any potential spelling errors.

Q2. Figure 7: Line 208, page 15. The binding of 10a was investigated. If so, please provide the Ka (of Kd). It will be good idea to give the Ka (Kd) of the control cyclic RGD peptide.

Response:

We thank the reviewer for the suggestions. To determine integrin $\alpha v \beta_3$ binding affinity of **10a** and **10c**, surface plasmon resonance (SPR) experiments were performed. Initially, the integrin was immobilized on a CM 5 chip using the amine coupling method. The chip was subsequently employed to measure the binding parameters of **10a** and commercially available **10c** cyclo-(RGDfK (FITC)) (analyte) over a range of concentrations from 3 to 100 μM and 6 to 200 μM , respectively. As shown in **Figure 7B**, the response units (RU) values for the binding of the cyclic peptide **10a** to integrin $\alpha v \beta_3$ exhibit a significant concentration dependence, with the determined equilibrium dissociation constant (K_D) of 6.75×10^{-7} M being lower than the K_D value of 6.55×10^{-5} M for the control peptide **10c**. K_a (K_d) for **10a** and **10c** are provided in **Table S2** highlighted in yellow on page S18 of Supplementary Information. Corresponding results and discussion are added on page 15-17 highlighted with yellow color.

Figure 7. A) The structures of cyclic RGD peptides **10a** and **10c** and fluorescent probes **10b** with thiol groups. **B)** SPR sensorgrams characterizing the affinity of peptides **10a** and **10c** to integrins targets.

Table S2. Equilibrium Dissociation Constant (K_D) Values of Cyclopeptides as Determined by the SPR Assay.

integrin	peptide	K_D (M)	k_a (1/Ms)	k_d (1/s)
$\alpha v \beta_3$	10a	6.75E-7	42325.156	0.028
$\alpha_5 \beta_1$	10a	8.36E-6	15322.862	0.128
$\alpha v \beta_5$	10a	6.13E-6	12523.125	0.077
$\alpha v \beta_3$	10c	6.55E-5	10251.531	0.671
$\alpha_5 \beta_1$	10c	2.76E-6	25354.351	0.069
$\alpha v \beta_5$	10c	4.13E-6	23357.393	0.096

*Q3. RGD peptides, including cyclic peptides bind to many integrins. Since the cycle is different than the control cyclicRGD, it will be an excellent idea to verify that **10a** has similar selectivity than the control for different integrins.*

Response:

We thank the reviewer for the suggestion. To verify the selectivity of **10a** and control **10c** for different integrins, surface plasmon resonance (SPR) experiments are conducted. We selected integrin receptors, including $\alpha v \beta_3$, $\alpha v \beta_5$, and $\alpha_5 \beta_1$, which are of clinical interest in anti-cancer therapy due to their expressions on the surface of cancer

cells and tumor neovasculature, as well as their role in mediating angiogenesis, tumor growth, and metastasis. K_D is the dissociation constant, which reflects the affinity of the compound to the target. As shown in **Figure 7B** and **Table S2**, the RU values for the binding of the cyclic peptide **10a** and **10c** to different integrin exhibit significant concentration dependence. **Table S2** displays the K_D , K_a , and K_d values of cyclic peptides **10a** and **10c** bound by different integrins in the SPR assay, indicating that **10a** exhibits higher affinity and selectivity than the control cyclic peptide **10c**. The results and discussion are shown on page 15-17 highlighted with yellow color.

Figure 7. A) The structures of cyclic RGD peptides **10a** and **10c** and fluorescent probes **10b** with thiol groups. B) SPR sensorgrams characterizing the affinity of peptides **10a** and **10c** to integrins targets.

Table S2. Equilibrium Dissociation Constant (K_D) Values of Cyclopeptides as Determined by the SPR Assay.

integrin	peptide	K_D (M)	k_a (1/Ms)	k_d (1/s)
$\alpha v \beta_3$	10a	6.75E-7	42325.156	0.028
$\alpha_5 \beta_1$	10a	8.36E-6	15322.862	0.128
$\alpha v \beta_5$	10a	6.13E-6	12523.125	0.077
$\alpha v \beta_3$	10c	6.55E-5	10251.531	0.671
$\alpha_5 \beta_1$	10c	2.76E-6	25354.351	0.069
$\alpha v \beta_5$	10c	4.13E-6	23357.393	0.096

Q4. HUVECs are the cells that are generally used for $\alpha v\beta_3$ testing. Please measure/give the number of $\alpha v\beta_3$ on the A549 cells.

Response:

We are grateful for the reviewer's suggestion. We designed a flow cytometry assay with HUVEC cells as reference to determine the amount of $\alpha v\beta_3$ in HeLa and A549 cells. As shown in **Figure 7C**, HeLa and A549 cells were 0.25 and 1.24-fold of HUVEC cells, respectively. The results and discussion are shown on page 17 highlighted in yellow.

Figure 7. C) Flow cytometry assay determining the number of $\alpha v\beta_3$ in HeLa and A549 cells. Living cells were pretreated with 5 μ M cyclic peptides **10a** for 60 min, and then with 5 μ M **10b** for 15 min. HUVEC cells were served as a reference. Data are expressed as mean \pm SD (experiment times n = 3). *p < 0.05, **p < 0.01, ***p < 0.001. n.s., not significant.

Q5. The manuscript is about a new chemical methodology. 1) Discussion should mention advantages and disadvantages of this methodology versus other ones. 2) What about the compatibility with peptides containing cysteines? 3) How to remove the reactants? 4) How toxic they are?

Response:

We thank the reviewer for the valuable comments.

1) We would like to highlight that some contributions of C-H functionalization on tryptophan have been made during the past decade by Lavilla/Albericio,^{1, 2} Ackermann,³⁻⁷ and Wang,⁸ among others.⁹⁻¹⁵ For example, Liu et al.¹³ successfully introduced maleimide into tryptophan and tryptophan-containing peptides at C(2) position by utilizing **N-pyridine** as directing group. Notably, most of them focused on the C(2) site owing to its high electron density, resulting in difficult to selectively C-H functionalize on the more inert C(4) and C(7) sites. Up to now, only limited examples of C(4)-H activation of tryptophan were achieved successfully assisted by directing groups.^{8, 16-18}

For Trp C(7) functionalization, Ackermann et al.¹⁹ recently reported a direct late-stage peptide C-H amidation at the tryptophan C(7) position by taking the N-pyridinyl group as directing group. However, they did not give the cyclization examples in this literature. Normally, to construct macrocyclic peptides, prefunctionalization, such as I, OTf, BR₂ and SnR₃, at C(7) position is essential.²⁰⁻²² To realize direct and site-selective cyclization via C(7)-H activation, we developed a Rh-catalyzed C(7) maleimidation and submitted the manuscript to Nat. Commun. on April 2022. Nearly at the same time, Wang and Shi²³ reported the Rh-catalyzed late-stage peptide ligation and

macrocyclization of C(7)-H alkylation of Trp with alkenes, utilizing the **N-P^tBu₂** directing group on May 2022. However, these reactions are different. The maleimidated peptides reported in this manuscript maintains the double bond. This is a biorthogonal group, and could rapidly couple with thiol containing biomolecules, which have potentially wide applications in chemical biology and drug discovery. Wang's cyclic peptides were achieved by addition reactions, which lost the possibility for further modifications. Thus, by taking advantages of this methodology, a peptide drug conjugation (PDC) **RGD-GFLG-DOX** by click chemical reaction between maleimide-cyclopeptide **10a** and thiol compound **15** was successfully constructed, which exhibited excellent specificity for tumor cells.

Figure S2. General Procedure for Removal of Directing and Protecting Groups in Functionalized Peptides

Additionally, the removal of these directing and protecting groups requires multiple and time-consuming steps (**Figure S2A, B**). To address this issue, we introduced N-pivaloyl as directing group for maleimidation on C(7)-H tryptophan and further investigated the strategy for removing the directing groups. Although no desired product was observed under traditional deprotecting conditions,²⁴⁻²⁶ we successfully found that trifluoroacetic acid could remove the directing group and protecting group

of the peptide at an ambient condition after extensive screening the conditions, offering the unprotected maleimide cyclic peptide (**Figure S2C**). We believe this finding will facilitate the further application of our method in the field of chemical biology, especially the construction of PDC.

Taken together, in contrast to all the highlighted previous works, we have in this study developed a novel, robust and regioselective rhodium-catalyzed methodology for late-stage diversification of Trp/Trp-containing peptides at the C(7) position, which exhibited significant application value in the fields of biochemistry, medicinal chemistry, and proteomics research. We have added these comments on page 4 and 13 highlighted in yellow.

2)

Figure 3. Substrate scope for maleimidation of dipeptides and complex peptides. Reaction conditions: **4** (0.2 mmol), **2a** (0.6 mmol), [RhCp*Cl₂]₂ (5 mol%), AgNTf₂ (20 mol%), Ag₂O (1.5 eq), DCM (1.5-2 mL), 80°C, 12 h.

As shown in **Figure 3** and **Figure 4**, we have synthesized peptides containing cysteine to verify the compatibility of the methodology. However, no desired product was generated in this method, possibly since cysteine is prone to oxidation. In addition, the unprotected active side chains of lysine-containing peptides (**4k**) will undergo Michael addition with maleimide, resulting in non-target products (**5k'**). More different peptides containing side chains of unprotected active amino acids were synthesized to verify the compatibility of the method with unprotected active amino acids. Fortunately, the results showed that the method was compatible with the unprotected active amino acids Asp (**5i**), Asn (**5j₁**), Gln (**5j₂** and **7k**), Ser (**7b**) and Thr (**7f**). The NMR and HRMS were included in SI. Results and discussion are added on page 8 and 11 highlighted in yellow.

Figure 4. Synthesis of complex peptides and hybrid molecules through C(7)-H activation chemical ligation. Reaction conditions: **1** or **4** (0.2 mmol), **6** (0.6 mmol), [RhCp*Cl₂]₂ (5 mol%), AgNTf₂ (20 mol%), Ag₂O (1.5 eq), DCM (1.5 mL), 80°C, 6-12 h

3) The method of removing the reactants is described in detail in this method and marked in yellow (on page 20 and Supporting Information page S8-S9). After cooling

to ambient temperature, diluted with DCM and passed through a short celite pad, the solvent was evaporated in vacuum to get the crude product, which was then further purified by flash column or PTLC.

4) The obtained cyclic peptides were subjected to corresponding cytotoxicity experiments, as shown in **Figure S4** with yellow highlighting, and most of these cyclic peptides exhibited no or low toxicity. The results and discussion are shown on page 13 highlighted in yellow.

Figure S4. Cytotoxicity of cyclic peptides on A549, U87MG, MCF-7, HeLa, LO2 and MIHA cells. Different cells were incubated with various concentrations of cyclic peptides (0-40 μM). Data are given as mean \pm sd ($n = 3$). After adding drugs, the cells were further incubated for another 36h.

References

1. Mendive-Tapia, L. et al. Constrained Cyclopeptides: Biaryl Formation through Pd-Catalyzed C-H Activation in Peptides-Structural Control of the Cyclization vs. Cyclodimerization Outcome. *Chem. – Eur. J.* **22**, 13114-13119 (2016).

2. Mendive-Tapia, L. et al. New peptide architectures through C-H activation stapling between tryptophan-phenylalanine/tyrosine residues. *Nat. Commun.* **6**, 7160 (2015).
3. Kaplaneris, N. et al. Late-stage stitching enabled by manganese-catalyzed C-H activation: Peptide ligation and access to cyclopeptides. *Sci. Adv.* **7**, eabe6202 (2021).
4. Kaplaneris, N. et al. Chemodivergent manganese-catalyzed C-H activation: modular synthesis of fluorogenic probes. *Nat. Commun.* **12**, 3389 (2021).
5. Lorion, M.M., Kaplaneris, N., Son, J., Kuniyil, R. & Ackermann, L. Late-Stage Peptide Diversification through Cobalt-Catalyzed C-H Activation: Sequential Multicatalysis for Stapled Peptides. *Angew. Chem. Int. Ed.* **58**, 1684-1688 (2019).
6. Ruan, Z., Sauermann, N., Manoni, E. & Ackermann, L. Manganese-Catalyzed C-H Alkynylation: Expedient Peptide Synthesis and Modification. *Angew. Chem. Int. Ed.* **56**, 3172-3176 (2017).
7. Schischko, A. et al. Late-stage peptide C-H alkylation for bioorthogonal C-H activation featuring solid phase peptide synthesis. *Nat. Commun.* **10**, 3553 (2019).
8. Bai, Z., Cai, C., Sheng, W., Ren, Y. & Wang, H. Late-Stage Peptide Macrocyclization by Palladium-Catalyzed Site-Selective C-H Olefination of Tryptophan. *Angew. Chem. Int. Ed.* **59**, 14686-14692 (2020).
9. Dong, H., Limberakis, C., Liras, S., Price, D. & James, K. Peptidic macrocyclization via palladium-catalyzed chemoselective indole C-2 arylation. *Chem. Commun.* **48**, 11644-11646 (2012).
10. Kee, C.W. et al. 18F-Trifluoromethanesulfinate Enables Direct C-H 18F-Trifluoromethylation of Native Aromatic Residues in Peptides. *J. Am. Chem. Soc.* **142**, 1180-1185 (2020).
11. Liu, J. et al. A Peptide Stapling Strategy with Built-In Fluorescence by Direct Late-Stage C(sp²)-H Olefination of Tryptophan. *Chem. – Eur. J.* **26**, 16122-16128 (2020).
12. Liu, J., Wang, P., Zeng, W., Lu, Q. & Zhu, Q. Late-stage construction of stapled peptides through Fujiwara-Moritani reaction between tryptophan and olefins. *Chem. Commun. (Camb)* **57**, 11661-11664 (2021).
13. Peng, J., Li, C., Khamrakulov, M., Wang, J. & Liu, H. Rhodium(III)-Catalyzed C-H Alkenylation: Access to Maleimide-Decorated Tryptophan and Tryptophan-Containing Peptides. *Org. Lett.* **22**, 1535-1541 (2020).
14. Reay, A.J. et al. Mild and Regioselective Pd(OAc)₂-Catalyzed C-H Arylation of Tryptophans by [ArN₂]X, Promoted by Tosic Acid. *ACS Catal.* **7**, 5174-5179 (2017).
15. Terrey, M.J., Holmes, A., Perry, C.C. & Cross, W.B. C-H Olefination of Tryptophan Residues in Peptides: Control of Residue Selectivity and Peptide-Amino Acid Cross-linking. *Org. Lett.* **21**, 7902-7907 (2019).
16. Li, J.J., Mei, T.S. & Yu, J.Q. Synthesis of indolines and tetrahydroisoquinolines from arylethylamines by Pd(II)-catalyzed C-H activation reactions. *Angew. Chem. Int. Ed.* **47**, 6452-6455 (2008).
17. Liu, Q., Li, Q., Ma, Y. & Jia, Y. Direct Olefination at the C-4 Position of Tryptophan via C-H Activation: Application to Biomimetic Synthesis of Clavicipitic Acid. *Org. Lett.* **15**, 4528-4531 (2013).
18. Zhang, Q. et al. Direct C4-Acetoxylation of Tryptophan and Tryptophan-Containing Peptides via Palladium(II)-Catalyzed C-H Activation. *Org. Lett.* **23**,

- 4699-4704 (2021).
19. Wang, W. et al. Peptide Late-Stage Diversifications by Rhodium-Catalyzed Tryptophan C7 Amidation. *Chem* **6**, 3428-3439 (2020).
 20. Jia, Y., Bois-Choussy, M. & Zhu, J. Synthesis of Diastereomers of Complestatin and Chloropeptin I: Substrate-Dependent Atropstereoselectivity of the Intramolecular Suzuki–Miyaura Reaction. *Angew. Chem. Int. Ed.* **47**, 4167-4172 (2008).
 21. Isley, N.A. et al. Total Synthesis and Stereochemical Assignment of Streptide. *J. Am. Chem. Soc.* **141**, 17361-17369 (2019).
 22. Deng, H. et al. Total Synthesis of Anti-HIV Agent Chloropeptin I. *J. Am. Chem. Soc.* **125**, 9032-9034 (2003).
 23. Liu, L. et al. P(III) -Directed Late-Stage Ligation and Macrocyclization of Peptides with Olefins by Rhodium Catalysis. *Angew. Chem. Int. Ed.* **61**, e202206177 (2022).
 24. Ruiz, M., Sánchez, J.D., López-Alvarado, P. & Menéndez, J.C. A systematic study of two complementary protocols allowing the general, mild and efficient deprotection of N-pivaloylindoles. *Tetrahedron* **68**, 705-710 (2012).
 25. Choi, I., Messinis, A.M. & Ackermann, L. C7-Indole Amidations and Alkenylations by Ruthenium(II) Catalysis. *Angew. Chem. Int. Ed.* **59**, 12534-12540 (2020).
 26. Behloul, C., Chouti, A., Guijarro, D., Nájera, C. & Yus, M. Reductive Removal of the Pivaloyl Protecting Group from Tetrazoles by a Naphthalene-Catalyzed Lithiation Process. *Synthesis* **47**, 507-510 (2014).

REVIEWERS' COMMENTS

Reviewer #1 (Remarks to the Author):

This revised manuscript addressed most of the concerns from this reviewer. However, this reviewer noticed that a paper was published on Organic Letters by Wang and Liu et al., reporting the development of an almost identical methodology for peptide modification (Org. Lett. 2023, 25, 2456–2460). This poses significant challenge to the novelty of this manuscript.

Reviewer #3 (Remarks to the Author):

The authors answered to the remarks. The new manuscript reports now a comprehensive investigation and a proof of concept that the newly formed cyclic peptides are good alternatives of formerly prepared ones.

RESPONSES TO REVIEWERS'

Reviewer #1 (Remarks to the Author):

This revised manuscript addressed most of the concerns from this reviewer. However, this reviewer noticed that a paper was published on Organic Letters by Wang and Liu et al., reporting the development of an almost identical methodology for peptide modification (Org. Lett. 2023, 25, 2456–2460). This poses significant challenge to the novelty of this manuscript.

Response:

We thank the reviewer for the comments and acknowledge the concern raised. Our manuscript was submitted to Nature Communications in April 2022 and the preprint was posted on Research Square in November 2022. In addition, we applied for a related patent titled “Maleimidated derivative of tryptophan-containing polypeptide and preparation method and medical use” in March 2022, and were published in June 2022. All details could be easily obtained through Scifinder. In February 2023, while our manuscript was under revision, Wang and Liu submitted a similar peptide modification work to Organic Letters and published in March 2023. Even so, our method still has several advantages over their method. Firstly, our method capitalizes on a lower temperature (80°C vs 120°C) and neutral solvent (dichloromethane vs hexafluoroisopropanol). Moreover, our method is compatible with unprotected amino acids, such as aspartic acid, asparagine, glutamine, serine, and threonine. Furthermore, we successfully cyclized dipeptides to hexapeptides, obtaining 22-40 membered cyclic peptides, including cyclic peptide **9j** with an RGD sequence. We also investigated the removal of directing group and found that the protecting and directing groups of peptide **9j** could be simultaneously removed in a one-step trifluoroacetic acid cleavage, resulting in cyclic peptide **10a**. Interestingly, maleimide-substituted tryptophan was demonstrated to have potential for thiol-ene click reactions. As an example, a peptide drug conjugation (PDC) **RGD-GFLG-DOX** by click chemical reaction between maleimide-cyclopeptide **10a** and thiol compound **15** was successfully constructed efficiently.

Taken together, our manuscript has demonstrated the novelty in the methodology and applications for the construction of PDC.

Reviewer #3 (Remarks to the Author):

The authors answered to the remarks. The new manuscript reports now a comprehensive investigation and a proof of concept that the newly formed cyclic peptides are good alternatives of formerly prepared ones.

Response:

Thank you very much for the reviewer's recognition of our revised work.